# Regiospecific α-methylene functionalisation of tertiary amines with alkynes via Au-catalysed concerted one-proton/two-electron transfer to O₂

Takafumi Yatabe [1] ✉ & Kazuya Yamaguchi [1] ✉

Regioselective transformations of tertiary amines, which are ubiquitously present in natural products and drugs, are important for the development of novel medicines. In particular, the oxidative α-C–H functionalisation of tertiary amines with nucleophiles via iminium cations is a promising approach because, theoretically, there is almost no limit to the type of amine and functionalisation. However, most of the reports on oxidative α-C–H functionalisations are limited to α-methyl-selective or non-selective reactions, despite the frequent appearance of α-methylene-substituted amines in pharmaceutical fields. Herein, we develop an unusual oxidative regiospecific α-methylene functionalisation of structurally diverse tertiary amines with alkynes to synthesise various propargylic amines using a catalyst comprising Zn salts and hydroxyapatite-supported Au nanoparticles. Thorough experimental investigations suggest that the unusual α-methylene regiospecificity is probably due to a concerted one-proton/two-electron transfer from amines to O₂ on the Au nanoparticle catalyst, which paves the way to other α-methylene-specific functionalisations.

Amines are ubiquitously present in nature and industry and play a pivotal role in organic chemistry[1]. In particular, tertiary amines are of great importance in the field of medicine; in fact, various tertiary amines exhibit biological activity and are utilised as drugs (Fig. 1a)[2–4]. For the development of novel compounds with medicinal properties, regioselective transformations such as late-stage C–H functionalisations of tertiary amines that preserve the structures of the parent amines are highly desired[5,6].

To date, the α-C–H functionalisation of tertiary amines has been actively studied[7–9] using various synthetic methods such as α-lithiation[10], directed α-C–H bond activations[11,12] and Rh-carbenoid insertions in α-C–H bonds[5,13]. Unfortunately, the application scope of these methods is intrinsically limited to certain amines and functionalisation. In contrast, oxidative α-C–H functionalisations can be

applied, theoretically, to an almost unlimited range of tertiary amines and functionalisations if the regioselectivity of the amine oxidation can be controlled. In fact, numerous catalytic oxidative α-C–H functionalisations with various nucleophiles via iminium cations[14–17] (or α-amino alkyl radicals)[18–20] have been developed (Fig. 1b); however, these processes generally occur in an α-methyl-selective manner because the typical mechanism of amine oxidation involves a single electron transfer (SET)/deprotonation/SET sequence (Path A, Supplementary Fig. 1), where the selectivity-determining step is the deprotonation of aminium radicals generated via SET from amines, which requires the half-vacant nitrogen p-orbital and the vicinal carbon p-orbital to overlap[21–23]. In other words, the regioselectivity switch from an α-methyl position to another position is difficult. Similarly, the regioselectivity of the amine oxidation cannot be controlled in other

[1]Department of Applied Chemistry, School of Engineering, The University of Tokyo, 7-3-1 Hongo, Bunkyo-ku, Tokyo 113-8656, Japan.
✉ e-mail: yatabe@appchem.t.u-tokyo.ac.jp; kyama@appchem.t.u-tokyo.ac.jp

**Fig. 1 | Overview of the oxidative α-C–H functionalisation of tertiary amines.** **a** Representative drugs and natural products possessing tertiary amine structures. **b** Difficulty in controlling the regioselectivity of the oxidative α-C–H functionalisation of tertiary amines. **c** Rare examples of oxidative α-C–H functionalisations showing exceptional regioselectivity to α-methylene C–H bond: cyanation and oxygenation. **d** This work: α-Methylene-specific alkynylation of tertiary amines via concerted one-proton/two-electron transfer to O₂ enabled by a combined catalyst of Au/HAP and ZnBr₂. THIQ tetrahydroisoquinoline, PG protecting groups, SET single electron transfer, HAP hydroxyapatite.

reported mechanisms such as hydrogen atom transfer (HAT)/SET (Path B)[24,25], SET/HAT (Path C)[26,27] and the Polonovski–Potier reaction via amine oxide formation (Path D)[28] (Supplementary Fig. 1), partly because α-C–H bonds of tertiary amines often possess nearly similar bond dissociation energies (e.g. 1-methylpiperidine: 92 kcal/mol for methyl *vs* 91 kcal/mol for methylene)[29,30]. In particular, the substrate scope for oxidative α-methylene functionalisations is generally limited to *N*-substituted benzylic amines like tetrahydroisoquinolines (THIQs), *N*-protected amines and symmetric amines (Fig. 1b)[7–9,14–20]. Considering the frequent appearance of α-methylene-substituted amines, especially cyclic methylene-substituted ones, in pharmaceutical fields[2–4,31], the development of an oxidative regioselective α-methylene C–H functionalisation is highly desirable.

Exceptionally, there are rare examples of oxidative α-C–H functionalisations showing regioselectivity to an α-methylene C–H bond: cyanation and oxygenation (Fig. 1c). In the case of cyanation, α-methyl-cyanated products can be transformed into thermodynamically stable α-methylene-cyanated products via azomethine ylides under cyanation conditions[32–36]. The α-methylene selectivity is dependent on the formation of azomethine ylides from the cyanated amines, which renders this method inapplicable to the other oxidative α-C–H functionalisations. In the case of the α-oxygenation of tertiary amines to produce amides, the selectivity toward α-methylene-oxygenation has been realised using (super)stoichiometric amounts of oxidants[37–39]. In this context, in our previous work, we successfully developed α-methylene-selective oxygenation of tertiary amines catalysed by Al₂O₃-supported Au nanoparticles;[40] however, the reason behind the observed regioselectivity could not be clarified. In addition, some examples of tertiary amine oxidations via β-hydride elimination exhibiting non-α-methyl selectivity (Path E, Supplementary Fig. 1) have been reported[41–43], but these systems have been hardly utilised for the α-functionalisation of tertiary amines. As a rare example, Beller et al. reported a Ru-catalysed α-methylene alkynylation of tertiary amines via β-hydride elimination[44]; however, the substrate scope was very narrow, and hydrogenation of the products usually occurred as a side reaction. In addition, during the revision of this paper, Slowing et al. also reported α-methylene-oxygenation reactions and a few limited examples of α-methylene alkynylation of tertiary amines via β-hydride elimination in the presence of an Au nanoparticle catalyst supported on mesoporous silica with pyridyl groups, while the yields (8–16%) and

turnover numbers (2–4) were quite low and insufficient for organic synthetic applications, and the detailed reaction mechanism for the amine oxidation was unrevealed[45]. Quite recently, Schoenebeck and Rovis et al. realised regioselective oxidative α-C–H alkylation of tertiary amines at the more-substituted positions by utilising the Curtin–Hammett principle via reversible and fast HAT catalysis;[30] however, in principle, the unique regioselectivity is derived not from the amine oxidation step but from the reaction between amino alkyl radicals and electrophiles, which limits the types of functionalisation to reactions like Giese radical addition, and regioselectivity control between sterically hindered positions (e.g., linear methylene vs cyclic methylene) is quite difficult[30]. In addition to the aforementioned reports, although a few reports exist on oxidative functionalisation systems showing very limited examples of α-cyclic methylene functionalisations[33,46–51], a wide range of tertiary amines, including unsymmetrical ones, cannot be utilised in the oxidative α-methylene selective functionalisation reactions developed to date (the previous main reports on oxidative α-methylene C–H functionalisations of tertiary amines except for benzylic amines, *N*-protected amines and symmetric amines are summarised in Supplementary Table 1). Considering this background, the development of novel general systems for the regioselective α-methylene functionalisation of tertiary amines containing a regioselective amine oxidation step would be an important breakthrough.

Herein, we develop an unusual oxidative regiospecific α-methylene C–H functionalisation of tertiary amines via iminium cation formation with alkynes that produces various propargylic amines, which are widely used in organic synthesis and the pharmaceutical fields[52], by utilising a combination of Zn salts and hydroxyapatite-supported Au nanoparticles (Au/HAP) as a catalytic system (Fig. 1d). The present catalytic system is applicable to a variety of aerobic α-methylene alkynylations of tertiary amines and affords propargylic amines, including cyclic derivatives except for THIQs and *N*-protected amines, which are difficult to synthesise using traditional methods[53–59]. Surprisingly, even in the presence of α-methine and linear-α-methylene C–H bonds, cyclic-α-methylene C–H bonds are regiospecifically alkynylated in this catalytic system. Thorough experimental investigations reveal that the unusual α-methylene regiospecificity probably arises from a unique amine oxidation mechanism: a concerted one-proton/two-electron transfer from the

**Table 1 | α-Methylene-specific alkynylation of 1-methylpiperidine (1a) with phenylacetylene (2a) using various catalysts and co-catalysts[a]**

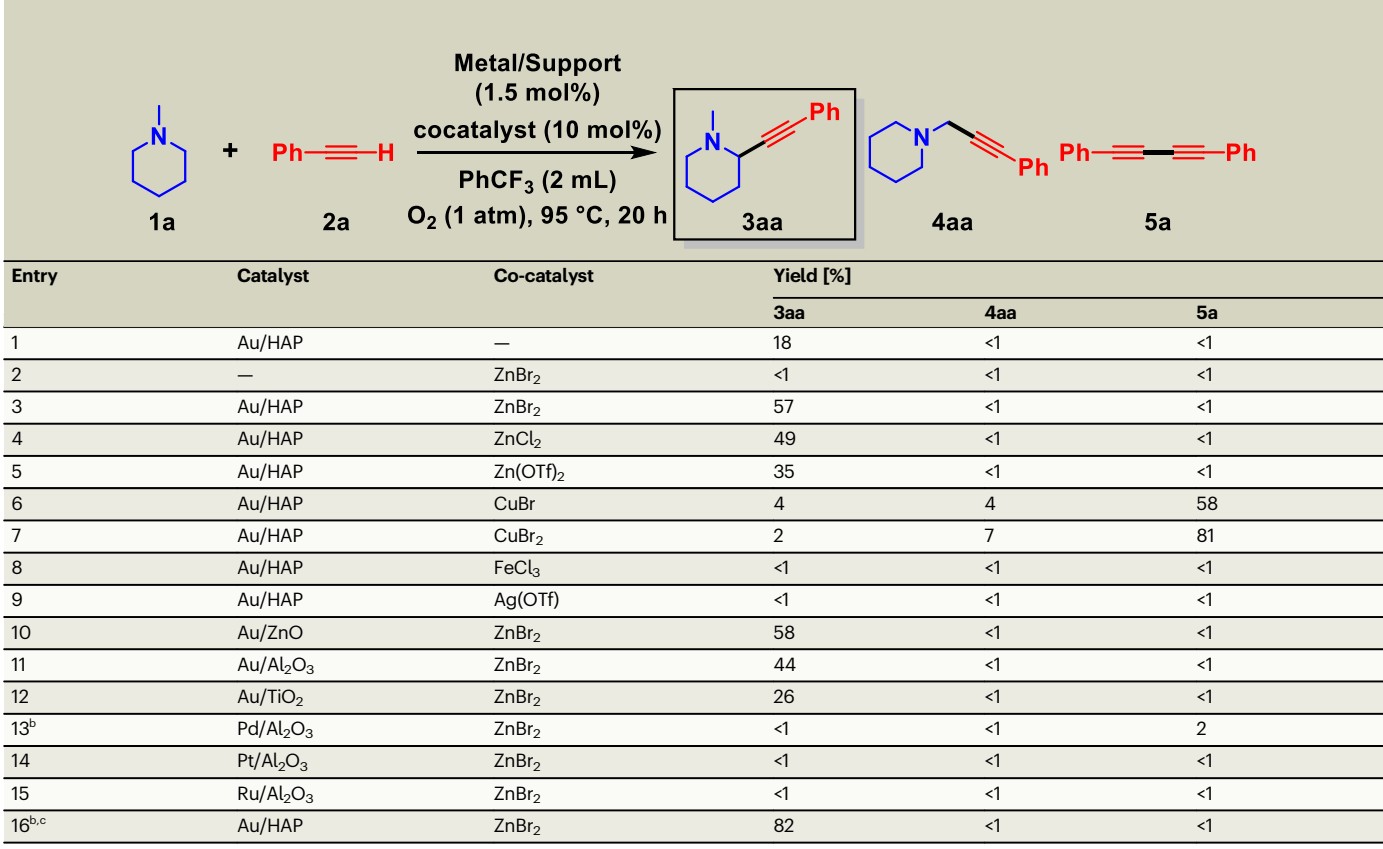

| Entry | Catalyst | Co-catalyst | Yield [%] | | |
|---|---|---|---|---|---|
| | | | 3aa | 4aa | 5a |
| 1 | Au/HAP | — | 18 | <1 | <1 |
| 2 | — | ZnBr₂ | <1 | <1 | <1 |
| 3 | Au/HAP | ZnBr₂ | 57 | <1 | <1 |
| 4 | Au/HAP | ZnCl₂ | 49 | <1 | <1 |
| 5 | Au/HAP | Zn(OTf)₂ | 35 | <1 | <1 |
| 6 | Au/HAP | CuBr | 4 | 4 | 58 |
| 7 | Au/HAP | CuBr₂ | 2 | 7 | 81 |
| 8 | Au/HAP | FeCl₃ | <1 | <1 | <1 |
| 9 | Au/HAP | Ag(OTf) | <1 | <1 | <1 |
| 10 | Au/ZnO | ZnBr₂ | 58 | <1 | <1 |
| 11 | Au/Al₂O₃ | ZnBr₂ | 44 | <1 | <1 |
| 12 | Au/TiO₂ | ZnBr₂ | 26 | <1 | <1 |
| 13[b] | Pd/Al₂O₃ | ZnBr₂ | <1 | <1 | 2 |
| 14 | Pt/Al₂O₃ | ZnBr₂ | <1 | <1 | <1 |
| 15 | Ru/Al₂O₃ | ZnBr₂ | <1 | <1 | <1 |
| 16[b,c] | Au/HAP | ZnBr₂ | 82 | <1 | <1 |

*1a* 1-methylpiperidine, *2a* phenylacetylene, *3aa* 1-methyl-2-(phenylethynyl)piperidine, *4aa* 1-(3-phenylprop-2-yn-1-yl)piperidine, *5a* 1,4-diphenylbuta-1,3-diyne, *HAP* hydroxyapatite, *OTf* trifluoromethanesulfonate.

[a]Reaction conditions: **1a** (0.5 mmol), **2a** (0.5 mmol), catalyst (metal: 1.5 mol%), co-catalyst (10 mol%), PhCF₃ (2 mL), 95 °C, O₂ (1 atm), 20 h. Conversions and yields were determined by gas chromatography using biphenyl as an internal standard.

[b]24 h.

[c]The results are given as average values of eight runs. **1a** (1 mmol), toluene (2 mL).

amines to O₂ on the Au nanoparticle catalyst, which differs from the conventional catalytic amine oxidation mechanisms.

## Results and discussion

### Effects of catalysts

We started out the investigation by screening various catalysts for the α-methylene-selective alkynylation of 1-methylpiperidine (**1a**) with phenylacetylene (**2a**) in trifluorotoluene (PhCF₃) as a solvent (2 mL) at 95 °C under an O₂ atmosphere (Table 1). In the presence of Au/HAP, the desired α-methylene-alkynylated product, i.e. 1-methyl-2-(phenylethynyl)piperidine (**3aa**), was obtained regiospecifically. Although the corresponding regioisomer 1-(3-phenylprop-2-yn-1-yl)piperidine (**4aa**, the α-methyl-alkynylated product) was not produced, the yield of **3aa** was not satisfactory, presumably because of the low nucleophilicity of the alkynyl species (Table 1, entry 1). The addition of a catalytic amount of ZnBr₂ to the reaction solution led to a drastic improvement in the yield of **3aa** (Table 1, entry 3). The use of ZnBr₂ as the only catalyst did not produce any amount of **3aa** (Table 1, entry 2), indicating that ZnBr₂ probably functioned as a co-catalyst to promote the nucleophilic addition of alkynes to iminium cations produced in the Au-catalysed amine oxidation[56,60]. Among the Zn salts examined, ZnBr₂ yielded the best results (Table 1, entries 3–5 and Supplementary Table 2, entries 1–5). Other metal co-catalysts known to promote the nucleophilic addition of alkynes, such as CuBr, CuBr₂, FeCl₃ and Ag(OTf)[61–63], proved inadequate for the present alkynylation (Table 1, entries 6–9).

Redox-active species like copper salts were especially unsuitable, affording **4aa** and 1,4-diphenylbuta-1,3-diyne (**5a**) via a SET process (Table 1, entries 6 and 7). We also investigated various supported Au nanoparticle catalysts (Table 1, entries 10–12 and Supplementary Table 2, entries 6–8). All the catalysts except for Au/layered double hydroxide (LDH) selectively afforded **3aa** in moderate-to-high yields, and Au/HAP and Au/ZnO worked comparatively well. Differences in the Au nanoparticle size in the catalysts were revealed by transmission electron microscopy, e.g. the Au nanoparticles in Au/TiO₂ were smaller than those in Au/HAP (Supplementary Fig. 2). Moreover, the X-ray photo-electron spectroscopy results reported in Supplementary Fig. 3 suggest that the differences of Au electronic states were not correlated with this α-alkynylation activity. Considering these results, the weak acid–base properties[64] or the low surface areas of Au/HAP and Au/ZnO are probably responsible for the efficient alkynylation of amines without side reactions. In the presence of the other noble-metal catalysts supported on Al₂O₃, no **3aa** was produced (Table 1, entries 13–15). Overall, the combined Au/HAP and ZnBr₂ system proved the most efficient catalyst for the present unusual α-methylene-specific alkynylation of tertiary amines.

The choice of the reaction solvent was also a key factor; low-polarity solvents like PhCF₃ and toluene were suitable (Supplementary Table 3). Under the optimised conditions, the desired product **3aa** was obtained in 82% yield (Table 1, entry 16). A hot filtration test (Supplementary Fig. 4) and an inductively coupled plasma atomic

emission spectrometry (ICP-AES) analysis of the solution revealed the heterogeneous nature of the Au/HAP catalysis (see details of the leaching test in the Supplementary Information). Au/HAP could be reused over at least two cycles of 24 h without lowering the **3aa** yield, although the reaction rate decreased with the number of cycles (Supplementary Table 4 and Supplementary Fig. 5). A characterisation of the used Au/HAP (Supplementary Figs. 6–9) revealed an increase in the Au nanoparticle size and the attachment of Zn species onto Au/HAP (for details of the reuse test, see the Supplementary Information).

## Substrate scope

We examined the substrate scope of the present regiospecific α-alkynylation of tertiary amines. The combined catalytic system afforded a variety of propargylic amines (**3**) from tertiary amines (**1**) and alkynes (**2**) with α-methylene specificity. The propargylic amines thus obtained were easily isolated by simple column chromatography on silica gel, and the isolated yields are shown in Figs. 2 and 3. First, we conducted the α-alkynylation of 1-methylpiperidine (**1a**) with various alkynes (**2**) (Fig. 2), which proceeded with complete α-methylene specificity. As mentioned above, **3aa** was efficiently obtained from **1a** and **2a** in a 78% isolated yield. When 2-, 3- and 4-chloro-substituted phenylacetylenes and the bromo-substituted analogue were used as substrates, the corresponding propargylic amines (**3ab**–**3ae**) were produced without dehalogenation. The α-methylene-specific alkynylation also proceeded effectively to afford the **3af**–**3ak** products using 4-fluoro-, 4-methoxy-, 4-methyl-, 4-trifluoromethyl-, 4-cyano- and 4-nitro-substituted phenylacetylenes. 3-Ethynylthiophene produced the corresponding propargylic amine (**3al**), whereas 2-ethynylpyridine was inapplicable (**3aq**). Aliphatic alkynes also proved suitable, even including an enyne and a propargylic alcohol (**3am**–**3ao**). Unfortunately, silyl alkynes and tosyl alkynes could not be used with this α-alkynylation system (**3ap** and **3ar**).

Next, we evaluated the α-methylene alkynylation of structurally diverse tertiary amines with **2a** (Fig. 3), finding that the corresponding single regioisomers were successfully obtained in almost all cases. The α-alkynylation of 1-methylpyrrolidine and 1-methylazepane effectively proceeded in a methylene-specific manner (**3ba** and **3ca**). Surprisingly, when using 4-hydroxymethyl-1-methylpiperidine as the substrate, the *trans*-α-methylene-alkynylated product was stereoselectively obtained with the alcoholic hydroxy group intact (**3da**). Moreover, a substrate possessing a tetrahydroisoquinoline skeleton was applicable to the present regiospecific alkynylation at the α-benzylic site (**3ea**). The use of *N*-substituted piperidines as substrates produced the corresponding propargylic amines (**3fa**–**3ja**) in moderate-to-high yields with selectivity toward the cyclic-α-methylene, even in the presence of a linear-α-alkyl group such as ethyl, benzylic or methine. Only in the synthesis of **3fa**, the linear-α-methylene-alkynylated product was observed in ~2% gas chromatography (GC) yield (**4fa**). Notably, the alkynylation of a linear tertiary amine also proceeded in a methylene-specific manner (**3ka**), and a bulky amine like triethylamine was suitable (**3la**). On the other hand, α-methine alkynylation of *N*,*N*-dimethylcyclohexylamine (**1m**) was difficult because of the preferential hydrolysis of the corresponding iminium cation compared with nucleophilic addition of **2a** (Supplementary Table 5, entry 1). When the amount of **2a** or ZnBr₂ was increased, the yield of the desired propargylic amine (**3ma**) was improved due to the promotion of alkyne nucleophilic addition (Supplementary Table 5, entries 2–4). The addition of molecular sieves 4 A (MS-4A) to remove H₂O in the reaction drastically increased the **3ma** yield (Supplementary Table 5, entries 5–8), and **3ma** was obtained in 84% GC yield (64% isolated yield) under the optimised conditions using MS-4A (300 mg) (Fig. 3, Supplementary Table 5, entry 7). The conditions with MS-4A were also suitable for more difficult selective alkynylation reactions of the following several amine substrates. When using 3-chloro-*N*,*N*-diethylpropan-1-amine as

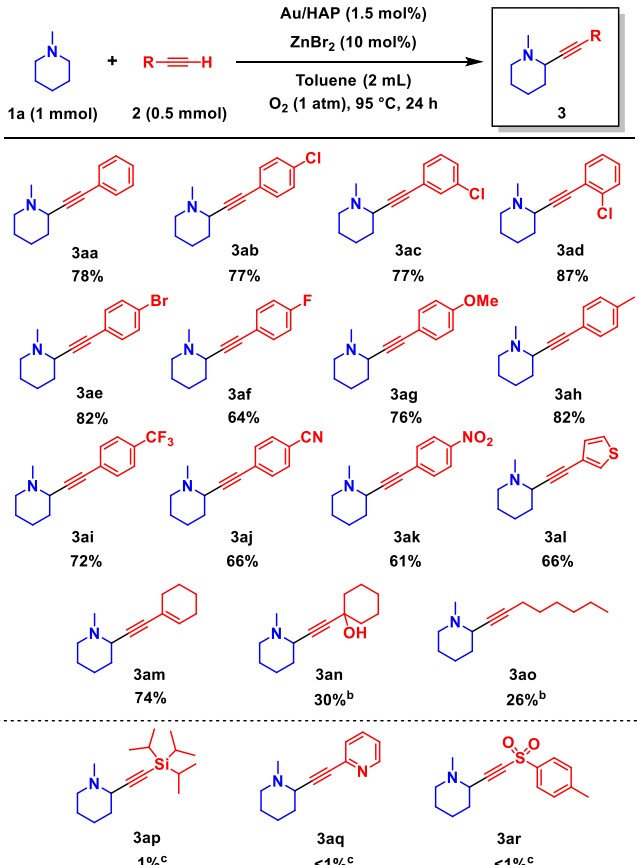

**Fig. 2 | Alkyne substrate scope of the combined catalytic system comprising Au/HAP and Zn species.** [a] [a]Reaction conditions: **1a** (1 mmol), **2** (0.5 mmol), Au/HAP (100 mg, Au: 1.5 mol%), ZnBr₂ (10 mol%), toluene (2 mL), 95 °C, O₂ (1 atm), 24 h. Isolated yields are shown. [b]ZnCl₂ (50 mol%), 36 h. [c]GC yields are shown. HAP hydroxyapatite.

the substrate, surprisingly, the α-methylene-alkynylated product at the ethyl group (**3na**) was regiospecifically obtained with the chloro group intact, suggesting that sterically non-hindered and/or electron-rich α-methylene C–H bonds can be selectively alkynylated among linear-α-methylene positions. Even in the presence of a cyclic-α-methine C–H bond with an ester group, cyclic-α-methylene selective alkynylation occurred to produce the corresponding *trans*-isomer stereoselectively (**3oa**). Likewise, the cyclic-α-methylene selective alkynylation of nicotine proceeded *trans*-stereoselectively (**3pa**), and as mentioned above, **3da** was also obtained as the *trans*-isomer stereoselectively. The *trans*-stereoselectivity is possibly derived from the attack of the alkynyl species from the opposite side of the catalyst surface to the iminium cation adsorbed on the Au nanoparticles. In addition, albeit in low yields, cyclic-α-methylene selective alkynylation of buflomedil and cloperastine successfully proceeded to afford propargylic amines **3qa** and **3ra**, demonstrating late-stage functionalisation of medicines possessing tertiary amine structures. Unfortunately, several amine substrates were not applicable to this alkynylation, as summarised in Supplementary Fig. 10. For example, as expected from the high methylene selectivity, *N*,*N*-dimethylaniline was not converted to the desired product. The inability of Au nanoparticles to catalyse amide oxidations[40] resulted in no reaction when using an amine protected by *tert*-butoxycarbonyl group. This reaction system was not applicable to the α-alkynylation of either quinuclidine or a secondary amine like piperidine. In the former case, the formation of the corresponding iminium cation would be forbidden by Bredt's rule, which supports the expected reaction pathway involving the production of iminium cation intermediates.

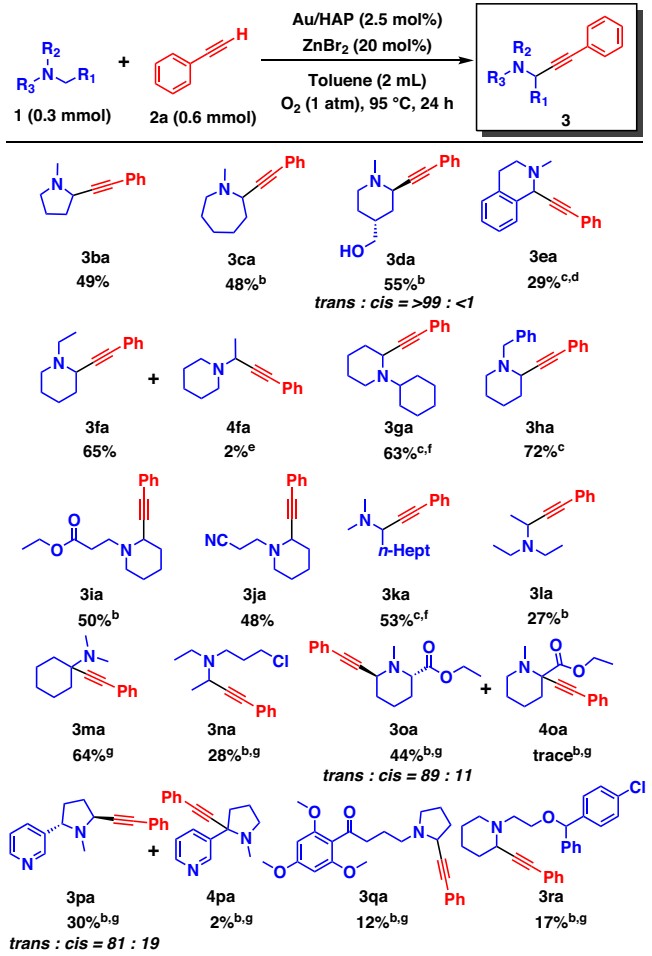

**Fig. 3 | Tertiary amine substrate scope of the combined catalytic system comprising Au/HAP and Zn species.** [a] [a]Reaction conditions: **1** (0.3 mmol), **2a** (0.6 mmol), Au/HAP (100 mg, Au: 2.5 mol%), ZnBr$_2$ (13 mg, 20 mol%), toluene (2 mL), 95 °C, O$_2$ (1 atm), 24 h. Isolated yields are shown. [b]Au/HAP (160 mg, 4 mol%). [c]**1** (1 mmol), **2a** (0.5 mmol), ZnBr$_2$ (11 mg, 10 mol%). [d]PhCF$_3$ (2 mL). [e]GC yield. [f]Au/HAP (200 mg). [g]**2a** (1.8 mmol), MS-4A (300 mg). The ratios of cis/trans-isomers were determined by [1]H NMR analysis or isolated yields. HAP hydroxyapatite.

## Mechanistic studies

According to previous reports on amine oxidation to iminium cations, five possible paths can be envisaged: path A, SET/deprotonation/SET; path B, HAT/SET; path C, SET/HAT; path D, Polonovski–Potier reaction via amine oxide formation; and path E, β-hydride elimination (Supplementary Fig. 1). To elucidate the mechanism of this unusual α-methylene-specific tertiary amine oxidation to iminium cations, we performed various control experiments. First, we investigated the involvement of radical species by adding a radical scavenger or using radical clocks as substrates. The addition of the radical scavenger 2,6-di-*tert*-butyl-4-methylphenol (BHT) under the present alkynylation conditions in an equimolar amount to **1a** did not affect the reaction rate (Supplementary Fig. 11). When using a radical clock as a substrate (**1s**), which could be expected to undergo ring opening upon oxidation via HAT and not to undergo ring opening by the iminium cation formation from density functional theory (DFT) calculations (Supplementary Figs. 12a and 13), for the alkynylation with **2a**, the corresponding alkynylated products (**3sa**) were obtained in moderate yields (diastereomeric ratio = 75:25) without formation of any ring-opened product (Fig. 4a). The diastereoselectivity is possibly due to the alkyne nucleophilic addition to the iminium cation on the Au nanoparticles as with the aforementioned *trans*-stereoselectivity in the case of **3da**, **3oa** and **3pa** or due to the steric effect of

phenylcyclopropyl group. Similarly, no ring opening reaction was observed when using **1t** as a radical clock, whose ring opening would proceed upon oxidation via SET (Supplementary Fig. 12b); the corresponding alkynylated product (**3ta**) was obtained in 6% yield, and deethylation of **1t** also occurred in 13% GC yield because it was difficult for the alkynyl species to attack the bulky iminium cation (Fig. 4a). These results suggest that the α-alkynylation reaction does not involve radical species. In addition, 4-methylmorpholine (**1u**) can be alkynylated with **2a** using the present catalytic system, whereas the reaction between 4-methylmorpholine *N*-oxide (**1u′**) and **2a** under the same conditions did not give the corresponding propargylic amine (**3ua**) at all (Supplementary Fig. 14), suggesting that this amine oxidation to iminium cations does not proceed via amine oxides. Next, we investigated the possible formation of iminium cations and Au–H species by β-hydride elimination on the Au nanoparticle catalyst using 1-methyl-piperidine-2,2-$d_2$ (**1a**-$d_2$) and treating **1a** with deuterated reagents (NaBD$_4$ or D$_2$O) in the presence of Au/HAP under an Ar atmosphere (Fig. 4b). In the presence of deuterated compounds, a hypothetical Au–H species would undergo H/D scrambling to produce the deuterated derivative of **1a** via an Au–D species. However, we found no clear evidence of such an H/D scrambling by GC-mass spectrometry (MS) in selected ion monitoring mode, [1]H nuclear magnetic resonance (NMR) and [13]C NMR (Supplementary Figs. 15–17), suggesting that a β-hydride elimination is not involved in the iminium cation formation. Therefore, it can be concluded that the present α-methylene-specific tertiary amine oxidation to iminium cations does not proceed via any of the aforementioned possible paths A–E.

Meanwhile, when the alkynylation reaction using Au/HAP and ZnBr$_2$ was conducted under an Ar atmosphere, a small amount of **2a** functioned as a hydrogen acceptor to produce an almost equal amount of **3aa** and styrene (**6a**) (Fig. 4c). The same result was observed in the absence of ZnBr$_2$, albeit with lower yields (Supplementary Fig. 18). Considering that Au–H species were not detected in the reaction without **2a** under an Ar atmosphere (Fig. 4b), a concerted one-proton/two-electron transfer possibly occurs from adsorbed **1a** to adsorbed **2a** on the Au nanoparticle catalyst under the conditions described in Fig. 4c. Thus, under an O$_2$ atmosphere, adsorbed O$_2$ species on Au nanoparticles could effectively function as the sole oxidant of an irreversible concerted one-proton/two-electron transfer from adsorbed **1a** to form iminium cations and an Au–OOH species, since the yield of **3aa** under an O$_2$ atmosphere was considerably higher than that under an Ar atmosphere (57 vs. 6% yield, respectively) and **6a** was not detected at all during the α-alkynylation under an O$_2$ atmosphere.

To gain more insight into this alkynylation, we performed a kinetic analysis of the reaction between **1a** and **2a** to produce **3aa** in the presence of Au/HAP (100 mg) and ZnBr$_2$ (0.05 mmol) (Supplementary Figs. 19–21). Although the initial production rate of **3aa** under air was independent of the initial concentration of **1a** or **2a**, it was clearly dependent on the partial pressure of O$_2$ (-0.6th order) (Fig. 4d). These results indicate that the turnover-limiting step is a reaction involving O$_2$ rather than amine adsorption or alkyne addition. Moreover, we examined the kinetic isotope effects (KIEs) using 1-methylpiperidine-2,2,6,6-$d_4$ (**1a**-$d_4$) or phenylacetylene-$d$ (**2a**-$d$). The intermolecular KIE between **1a** and **1a**-$d_4$ based on independently determined production rates was large ($k_H/k_D = 3.6$), revealing the α-methylene C–H bond cleavage of **1a** as the turnover-limiting step (Fig. 4e and Supplementary Fig. 22)[65]. The kinetic results show that the turnover-limiting step is the amine oxidation with O$_2$ on the Au nanoparticles, which is an evidence of the concerted one-proton/two-electron transfer from adsorbed amines to adsorbed O$_2$. Although one-proton/two-electron (hydride) transfer from tertiary amines to stoichiometric oxidants has been reported to date[58,66,67], to our knowledge, a concerted one-proton/two-electron transfer from tertiary amines to O$_2$ via a catalyst exhibiting α-methylene selectivity is hitherto unknown. By contrast, when **2a**-$d$ was

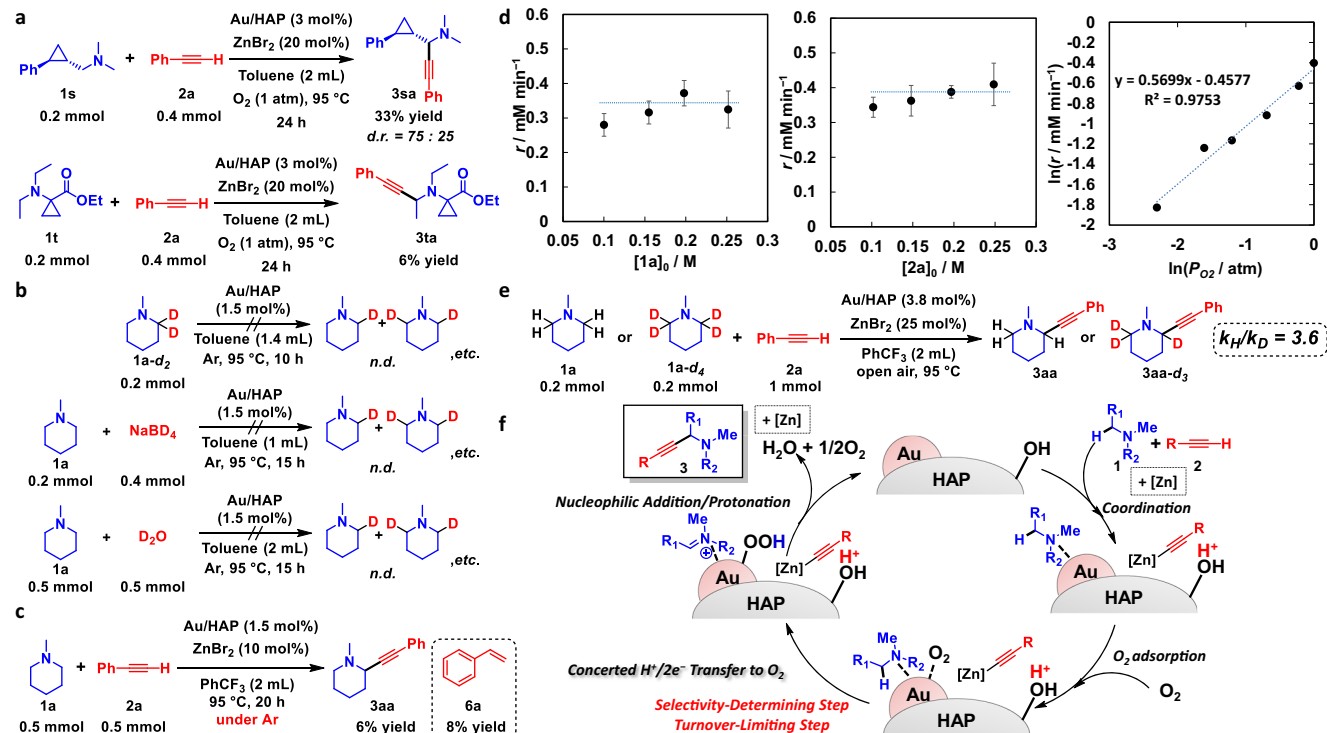

**Fig. 4 | Overview of the α-methylene-specific alkynylation mechanism.**
**a** Alkynylation of radical clocks (**1s** or **1t**) with **2a**. Isolated yields are shown.
**b** Scrambling of deuterium using **1a-d₂**, **1a** and NaBD₄ or **1a** and D₂O. **c** Alkynylation of **1a** with **2a** under Ar. GC yields are shown. **d** Dependence of the **3aa** production rate on the concentration of **1a**, the concentration of **2a** and the O₂ partial pressure.

The error bars calculated from two runs show the maximum/minimum values of the respective production rates of **3aa** ($r$). **e** Kinetic isotope effects on the alkynylation using **1a** or **1a-d₄** with **2a**. **f** Plausible mechanism of the α-methylene-specific alkynylation. d.r. diastereomeric ratio. n.d. not detected. HAP hydroxyapatite.

used as the substrate instead of **2a**, the production rate of **3aa** decreased a little ($k_H/k_D = 1.3$), although GC-MS patterns revealed the quick H/D exchange of **2a-d** (estimated deuteration ratio as of 10 min: ~17%) (Supplementary Figs. 23, 24). Considering the aforementioned kinetic analysis and the quick H/D exchange, this decrease in the **3aa** production rate was not probably derived from KIE of **2a** C–H cleavage. On the other hand, surprisingly, **3aa** was deuterated in ~62% as of 10 min after the reaction started using **2a-d** without any deuteration of **1a** (Supplementary Figs. 23, 25, 26). After the reaction for 24 h, the product was selectively isolated via column chromatography as 1-methyl-2-(phenylethynyl)piperidine-3,3-d₂ (**3aa-β-d₂**) (deuteration ratio: 13%). Thus, these results indicated that enamines formed from iminium cations accepted deuteron instead of proton to convert into deuterated iminium cations during the reaction (Supplementary Fig. 27). This is an evidence of iminium cation presence in the catalytic system, and irreversible concerted one-proton/two-electron transfer to O₂ form iminium cations was strongly supported by the regiospecifically β-deuterated **3aa** formation. Moreover, azomethine ylide formation[34,35,68] and isomerization of iminium cations[69] were excluded in this catalytic system. The deuteration ratio decrease of **3aa** after the reaction for 24 h compared with the initially formed **3aa** in the reaction is probably derived from the increase of proton source (e.g. H₂O) as the oxidation reaction proceeds. In other words, the addition of D₂O is assumed to give deuterated propargylic amines from non-deuterated amines and alkynes. In fact, in the presence of a large amount of D₂O (3 mmol, three equivalents to **1a**), selectively β-deuterated propargylic amine **3aa-β-d₂** (deuteration ratio: 76%) was successfully synthesised in 44% yield (Supplementary Fig. 28), which will be beneficial in the synthesis of deuterated medicines[70]. To reveal the reason for the decrease of **3aa** yield in the presence of D₂O, either H₂O or D₂O was added to the present alkynylation, revealing the inhibition effect of water on **3aa** production, possibly because of water adsorption on

Au nanoparticles (Supplementary Fig. 29). Furthermore, the $k_{H_2O}/k_{D_2O} = 1.5$ was observed without much deuteration of **2a** (~6–9%) and almost equivalent to that using **2a** or **2a-d** ($k_H/k_D = 1.3$). Thus, the regeneration step of Au–OOH species to Au species might be affected by the deuteration of **2a** or the addition of D₂O.

In addition, the intramolecular KIE using **1a-d₂** was determined by ¹H NMR ($k_H/k_D = 3.3$) to be almost equal to the intermolecular KIE (Supplementary Fig. 30), indicating that the present regioselectivity is determined by kinetic control at the concerted one-proton/two-electron transfer to O₂ rather than by the thermodynamic stability of the iminium cation intermediates. At the transition state of the concerted one-proton/two-electron transfer, an iminium cation-like species is probably formed on the Au nanoparticles. Thus, in some cases like the α-alkynylation of **1a** and 1-ethylpiperidine (**1f**), the regioselectivity is correlated to the stability differences of the corresponding iminium cations; however, the selectivity of other cases, such as the reaction of 1-cyclohexylpiperidine (**1g**) or 3-chloro-N,N-diethylpropan-1-amine (**1n**), is different from the stability order because of kinetic factors like steric effects and electronic effects (Supplementary Fig. 31), which probably leads to the unusual α-methylene selectivity.

According to the results, we propose the mechanism illustrated in Fig. 4f for this unusual α-methylene-specific alkynylation catalysed by the combination of Au/HAP and ZnBr₂. First, a tertiary amine coordinates an Au nanoparticle and alkyne coordinates to a Zn species via deprotonation. Subsequently, O₂ is adsorbed on the Au nanoparticle, followed by a concerted one-proton/two-electron transfer to produce the corresponding iminium cation and an Au–OOH species. This concerted amine oxidation step is both the selectivity-determining step and the turnover-limiting step. Then, the desired propargylic amine is produced via the nucleophilic addition of the alkyne to the iminium cation. The trans-stereoselectivity to alkynylation of substituted cyclic tertiary amines is thought to be derived from this step,

possibly due to the adsorption of the iminium cations on the Au nanoparticle. In fact, DFT calculation results using an $Au_{20}$ cluster model and the corresponding iminium cation of **1a** indicated that the iminium cation is adsorbed and stabilised on the Au nanoparticle (Supplementary Fig. 32). In addition, when the oxidation of **1a** in the absence of **2a** was carried out using Au/HAP with/without $ZnBr_2$, the conversion of **1a** to the corresponding iminium cation, enamine and amide was quite low; however, the presence of **2a** drastically improved the conversion of **1a**, especially with $ZnBr_2$ (Supplementary Table 6), suggesting that the nucleophilic addition of **2a** removed the iminium cations (and enamines) adsorbed on Au nanoparticles to increase the turnover number of amine oxidation. Thus, the nucleophilic addition of the alkyne promoted by the Zn species to the iminium cation adsorbed on the Au nanoparticle presumably affords the desired propargylic amine in this step. Finally, the Au–OOH species accepts a proton to afford Au, $H_2O$ and $O_2$, closing the catalytic cycle.

In summary, the combined catalyst system comprising Au/HAP and $ZnBr_2$ promotes an unusual α-methylene-specific alkynylation of tertiary amines, which probably proceeds via an irreversible concerted one-proton/two-electron transfer from amines to $O_2$ on the Au nanoparticle catalyst, differently from conventional amine oxidation mechanisms. In the presence of α-methine and linear-α-methylene C–H bonds, cyclic-α-methylene C–H bonds were regiospecifically alkynylated, and *trans*-stereoselective alkynylation was also observed in the case of substituted cyclic tertiary amines. Most of the as-produced propargylic amines exhibit unprecedented structures, including alkynylated buflomedil and cloperastine, demonstrating the great utility of this transformation to synthesise unexplored drug candidates and synthetic building blocks. This report provides a general example of α-methylene-specific oxidative C–H alkynylation of tertiary amines, which paves the way for other various α-methylene-specific functionalisations. We believe that this transformation will have a great influence on precise organic synthesis based on amine moieties and bridge the gap between heterogeneous catalysis and the development of novel reactions in organic chemistry.

## Methods

### Preparation of Au/HAP

Au nanoparticles supported on HAP (Au/HAP) were prepared as follows: HAP (2.0 g) was added to an aqueous solution of $HAuCl_4$ (2 mM, 100 mL). After vigorously stirring the mixture for 2 min, aqueous $NH_3$ (10%, 240 μL) was added, and the resulting mixture was stirred at room temperature for 14 h. The resulting slurry was filtered, washed with deionised water (1 L) and dried at room temperature in vacuo to give the HAP-supported Au precursor. The resulting species was dispersed in deionised water (100 mL) and treated with $NaBH_4$ (80 mg) at room temperature for 1 h. The mixture was then filtered, and the residue was washed with deionised water (1 L) and dried to afford the Au/HAP catalyst as a reddish-purple powder (Au content as determined by ICP-AES: 1.5 wt%). Following the same procedure using ZnO, $Al_2O_3$, $ZrO_2$, $CeO_2$, $TiO_2$ or LDH as the support instead of HAP, Au/ZnO (Au content: 1.5 wt%), Au/$Al_2O_3$ (Au content: 1.8 wt%), Au/$ZrO_2$ (Au content: 1.3 wt%), Au/$CeO_2$ (Au content: 1.6 wt%), Au/$TiO_2$ (Au content: 1.6 wt%) and Au/LDH (Au content: 1.4 wt%) were prepared.

### Catalytic reaction

The catalytic reaction under the optimised conditions was typically performed according to the following procedure: $ZnBr_2$ (10 mol%), Au/HAP (Au: 1.5 mol%), biphenyl (0.1 mmol, internal standard), 1-methylpiperidine (**1a**, 0.5 mmol), phenylacetylene (**2a**, 0.5 mmol), toluene (2 mL) and a Teflon-coated magnetic stir bar were successively placed into a Pyrex glass reactor (volume: ~20 mL). After purging with $O_2$ for 1 min, the mixture was stirred at 95 °C under a closed $O_2$ atmosphere (1 atm). The yields of the products were determined by GC

analysis using biphenyl as an internal standard. With respect to the isolation of products, after the reaction, the catalyst was removed by simple filtration, and the filtrate was concentrated by evaporation of toluene. The crude product was subjected to column chromatography on silica gel (typically using hexane/ethyl acetate = 6/4 or 8/2 as eluent) to produce the pure propargylic amines (respective eluents were shown in spectral data of Supplementary Information). The products were identified by GC-MS, NMR ($^1$H, $^{13}$C and $^{19}$F) and elemental analysis of C, H and N.

## Data availability

The data generated in this study are provided within the article and its Supplementary Information file. Correspondence and requests for materials should be addressed to K.Y. and T.Y.

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

## Acknowledgements

This work was financially supported by JSPS KAKENHI Grant No. 17J08127 (T.Y.), 19H02509 (K.Y.), 20K22547 (T.Y.), 21K14460 (T.Y.) and 22H04971 (K.Y. and T.Y.). A part of this work was conducted at the Advanced Characterisation Nanotechnology Platform of the University of Tokyo, supported by the Nanotechnology Platform of the Ministry of Education, Culture, Sports, Science and Technology (MEXT), Japan. Some of the computations were performed using Research Centre for Computational Science (RCCS), Okazaki, Japan. We thank Mr. Satoru Nakai for the synthesis of some of the *N*-methyl tertiary amines and for helpful discussions. We also thank Mr. Wei-Chen Lin for some experiments and synthesis for the responses to referees.

## Author contributions

T.Y. conceived, designed and mainly performed the experiments. K.Y. and T.Y. analysed the data and wrote the paper.

## Competing interests

The authors declare no competing interests.
