## [Peer Review File · Nature Communications]

Regiospecific α -methylene functionalisation of tertiary amines with alkynes via Au-catalysed concerted one-proton/two-electron transfer to O₂REVIEWER COMMENTS

Reviewer #1 (Remarks to the Author):

Yatabe and Yamaguchi report an unusual oxidative regioselective α -methylene functionalisation of tertiary amines with alkynes by using Zn salts and Au/HAP as catalyst and O₂ as oxidant. The manuscript first focused on the alkynylation of methyl-protected piperidine; electronically varied arylacetylenes were suitable nucleophiles with moderate to good yields; aliphatic acetylenes only gave 26% yield in one example; then other types of cyclic amines including pyrrolidine and 7-membered substrate were tested with moderated yields. A systematic mechanistic study was conducted, suggesting a concerted hydride transfer for iminium ion formation.

The reviewer's opinions are listed below:

First, oxidative C-H functionalization of tertiary amines is an established strategy for C-C bond formation, and has received considerable attentions. Besides cyanation and oxygenation, other types of reactions including arylation, alkylation, and alkenylation of saturated tertiary amines with similar regioselectivity have been disclosed (Chem. Eur. J. 2015, 21, 16272–16279; Org. Lett., 2007, 9, 24, 5115–5118; Eur. J. Org. Chem. 2017, 4188–4193; Chem. Commun., 2009, 3169–3171). Notably, C-H alkynylation of saturated tertiary amine are also known (Org. Chem. Front., 2018, 5, 3515–3519; Chem. Commun., 2010, 46, 1956–1958). Accordingly, the reviewer cannot agree with the statement of the novelty in the manuscript.

Second, the scope of the reaction is rather narrow with moderate efficiency. The work shares the routine scope of the above precedents. Whether the reaction can achieve excellent regio- and diastereoselectivities for unsymmetric tertiary amines? What about the asymmetric version of the transformation? Without these studies, the reviewer did not see the lightspots of the work to be published in Nat. Commun.

Third, while a thorough experimental investigation was conducted, the hydride transfer mechanism for oxidative C-H functionalization is known, which should be cited (ex. Org. Lett. 2016, 18, 6476–6479; Eur. J. Org. Chem. 2010, 4460–4467; J. Org. Chem. 2015, 80, 16, 8150–8167).

Based on the above three major points, I do not recommend publication of the manuscript in Nat. Commun. Other specific organic chemistry journal might be suitable for the manuscript.

Other comments:

- 1) The substrate scope is mainly focused on cyclic tertiary amines. More acyclic substrates having different kind of functional groups may be tested to show the substrate scope of the reaction further.
- 2) 3ka and 3la were obtained in 54% and 27% isolated yield. Are there any demethylated or dealkylated products generated?
- 3) Check the structure of compounds 3qa.
- 4) SI: Please report the yields of the final products and insert them before the description of the NMR spectra.
- 5) In the ¹³C NMR spectrum of 3ao, better signal: noise is needed to properly identify the quartet with the largest splitting.

Reviewer #2 (Remarks to the Author):

This manuscript is a sound contribution in the fields of Au nanoparticle catalysis and synthetic organic methodology as well. It is initiated by previous studies from the same group regarding the selective aerobic oxidation of amines into amides (ref 37), and herein it is shown that suitable acetylides can C-C couple on the thermodynamically most stable iminium cation, which has very few preceding examples in the literature. What is obscure and needs further attention is the mechanistic part.

a) The possibility of the generation of an iminium cation as an intermediate appears reasonable, as it fits nicely with the regioselectivity of the C-C coupling, except the case of 3ga where alkynylation is kinetically driven. Thus, in the case of N-methyl piperidine as an example, one would expect the formation of the most stable endocyclic iminium cation versus the exocyclic (Supplementary Fig. 23), as occurs. The authors have studied amines 1q and 1r as probes to exclude the formation of radical cations, as no ring opening rearrangement is observed in their alkynylations. However, I would anticipate the iminium cation from 1q to undergo ring opening, as α -cyclopropyl probes are sensitive not only to radical but to carbocations as well. My impression is that an iminium cation bound on Au nanoparticle is the most possible intermediate. Due to the low polarity of Au-C bonds (JACS Au, 2021, 1, 362) the intermediate might not be exactly ionic but rather having partial charges that do not possible allow ring opening rearrangement in the case of 1q. There is an additional example of none phenylcyclopropyl ring opening (Org. Lett. 2019, 21, 5552) presumably for the same reasons. I believe that the mechanistic analysis needs revision, and furthermore, theoretical calculations on a model nanoparticle may help to clarify the situation.

b) Among the isotope effect studies presented herein, I believe that the $k_H/k_D = 1.2$ presented Supplementary Fig. 20 may not be valid, as D-labelled terminal alkynes often undergo deuterium depletion in the presence of Au NPs. Additionally, I am quite skeptical regarding the information gained from the intermolecular KIE of $k_H/k_D = 3.6$, between 1a and 1a-d4 suggesting the α -methylene C-H bond cleavage the turnover-limiting step. As more possibly a stable iminium cation bound on Au NP is formed as an intermediate, I would anticipate a $k_H/k_D > 1$ regardless of the kinetic profile of the alkynylation step. Based on the analysis provided above, publication in Nature Chemistry is recommended after significant improvement of the mechanistic analysis.

Reviewer #3 (Remarks to the Author):

In my opinion, it is an excellent, well written paper. Everything is very well explained and clearly stated. It can be accepted as it is.

In terms of personal preference, I would prefer seeing some more figures in the paper, not have almost everything in SI, however that is a matter of the author's preference.

Point-by-Point Responses

<To Reviewer #1>

Comments

Yatabe and Yamaguchi report an unusual oxidative regiospecific α -methylene functionalisation of tertiary amines with alkynes by using Zn salts and Au/HAP as catalyst and O₂ as oxidant. The manuscript first focused on the alkynylation of methyl-protected piperidine; electronically varied arylacetylenes were suitable nucleophiles with moderate to good yields; aliphatic acetylenes only gave 26% yield in one example; then other types of cyclic amines including pyrrolidine and 7-membered substrate were tested with moderated yields. A systematic mechanistic study was conducted, suggesting a concerted hydride transfer for iminium ion formation.

The reviewer's opinions are listed below:

First, oxidative C-H functionalization of tertiary amines is an established strategy for C-C bond formation, and has received considerable attentions. Besides cyanation and oxygenation, other types of reactions including arylation, alkylation, and alkenylation of saturated tertiary amines with similar regioselectivity have been disclosed (Chem. Eur. J. 2015, 21,16272 –16279; Org. Lett., 2007, 9, 24, 5115–5118; Eur. J. Org. Chem. 2017, 4188–4193; Chem. Commun., 2009, 3169-3171). Notably, C-H alkynylation of saturated tertiary amine are also known (Org. Chem. Front., 2018, 5, 3515–3519; Chem. Commun., 2010, 46, 1956–1958). Accordingly, the reviewer cannot agree with the statement of the novelty in the manuscript.

Second, the scope of the reaction is rather narrow with moderate efficiency. The work shares the routine scope of the above precedents. Whether the reaction can achieve excellent regio- and diastereoselectivities for unsymmetric tertiary amines? What about the asymmetric version of the transformation? Without these studies, the reviewer did not see the lightspots of the work to be published in nat com.

Third, while a thorough experimental investigation was conducted, the hydride transfer mechanism for oxidative C-H functionalization is known, which should be cited (ex. Org. Lett. 2016, 18, 6476–6479; Eur. J. Org. Chem. 2010, 4460–4467; J. Org. Chem. 2015, 80, 16, 8150–8167).

Based on the above three major points, I do not recommend publication of the manuscript in Nat. Commun. Other specific organic chemistry journal might be suitable for the manuscript.

Responses

Thank you for your kind review. According to your valuable and helpful comments, we did various additional experiments and revised the manuscript and the Supplementary Information thoroughly, especially about the introduction, substrate scopes, and mechanistic studies. We believe that the revised version is much improved and sufficiently suitable for the publication in *Nature Communications*. Please confirm the following responses, the revised manuscript, and the revised SI.

Comments

First, oxidative C-H functionalization of tertiary amines is an established strategy for C-C bond formation, and has received considerable attentions. Besides cyanation and oxygenation, other types of reactions including arylation, alkylation, and alkenylation of saturated tertiary amines with similar regioselectivity have been disclosed (Chem. Eur. J. 2015, 21,16272 –16279; Org. Lett., 2007, 9, 24, 5115–5118; Eur. J. Org. Chem. 2017, 4188–4193; Chem. Commun., 2009, 3169-3171). Notably, C-H alkynylation of saturated tertiary amine are also known (Org. Chem. Front., 2018, 5, 3515–3519; Chem. Commun., 2010, 46, 1956–1958). Accordingly, the reviewer cannot agree with the statement of the novelty in the manuscript.

Responses

Thank you for introducing several reports on oxidative C–H functionalizations of tertiary amines. In the introduced reports, it is true that α -methylene functionalized tertiary amines except for *N*-substituted benzylic amines like tetrahydroisoquinolines (THIQs), *N*-protected amines, and symmetric amines were produced with α -cyclic methylene selectivity, but the substrate scopes of the α -methylene functionalization were quite limited (only 1–4 examples per one report), and in some cases, α -methyl C–H bonds were also oxidized and functionalized substantially, which indicated the non-regioselectivity to oxidative C–H functionalizations of tertiary amines (*Org. Lett.* **2007**, *9*, 5115–5118; *Eur. J. Org. Chem.* **2017**, 4188–4193). In addition, in all of the introduced reports, the regioselectivity to α -methyl vs α -linear methylene was not demonstrated, or α -methyl selective oxidative functionalization occurred in the presence of α -linear methylene sites (*Org. Lett.* **2007**, *9*, 5115–5118; *Eur. J. Org. Chem.* **2017**, 4188–4193; *Org. Chem. Front.* **2018**, *5*, 3515–3519). These non-selectivity and α -methyl selectivity were probably derived from the typical mechanism of amine oxidation involving a single

electron transfer (SET)/deprotonation/SET sequence, where the selectivity-determining step is the deprotonation of aminium radicals generated via SET from amines, which requires the half-vacant nitrogen p-orbital and the vicinal carbon p-orbital to overlap. Also, it is possibly because α -C–H bonds of tertiary amines often possess nearly similar bond dissociation energies (e.g. 1-methylpiperidine: 92 kcal/mol for methyl vs 91 kcal/mol for methylene). Although α -cyanation and oxygenation reactions, which we classified as the exception, demonstrated α -methyl functionalizations in some cases, the ratio of methyl-functionalized products was comparatively low, and the substrate scopes were broader than those of the introduced reports. When focusing on the previous reports on α -alkynylation of tertiary amines, in the case of the report from Shi's group (*Org. Chem. Front.* **2018**, 5, 3515–3519), α -methylene selective alkynylation in the presence of α -methyl C–H bonds could not be demonstrated (on the contrary, α -methyl selective alkynylation was demonstrated in the presence of α -linear methylene sites). Beller et al. reported an Ru-catalysed α -methylene alkynylation of tertiary amines via β -hydride elimination (*Chem. Commun.* **2010**, 46, 1956–1958); however, the substrate scope was very narrow (limitation of α -methylene selective functionalization to only four kinds of cyclic amines, limitation of alkynes to only silyl alkynes, and no demonstration of functional groups), and hydrogenation of the products usually occurred as a side reaction. Furthermore, in most of previously reported oxidative α -alkynylation reactions, the substrate scopes of α -methylene alkynylation were limited to *N*-substituted benzylic amines, *N*-protected amines, and symmetric amines (e.g., *J. Am. Chem. Soc.* **2004**, 126, 11810–11811; *J. Org. Chem.* **2008**, 73, 3961–3963; *Org. Lett.* **2009**, 11, 1027–1029; *RSC Adv.* **2014**, 4, 34712–34715; *Green Chem.* **2016**, 18, 3499–3502; *Org. Lett.* **2016**, 18, 6476–6479; *ACS Catal.* **2018**, 8, 10032–10035).

By contrast, the present oxidative C–H alkynylation of tertiary amines by utilizing a combination of ZnBr₂ and hydroxyapatite-supported Au nanoparticles (Au/HAP) is totally different from the previously reported oxidative C–H functionalizations of tertiary amines and exhibited the α -methylene regioselectivity to structurally diverse tertiary amines (19 examples including unsymmetric tertiary amines demonstrated in this revised version, see the responses to your second comments shown below) except for *N*-substituted benzylic amines, *N*-protected amines, and symmetric amines. First of all, this catalytic system does not functionalize α -methyl C–H bonds of tertiary amines at all even with linear α -methylene C–H bonds (e.g., *N,N*-dimethyloctylamine) or linear α -methine C–H bonds (e.g., *N,N*-dimethylcyclohexylamine) but regioselectively alkynylated α -methylene C–H bonds. In the case of *N,N*-dimethylcyclohexylamine, possessing no α -methylene C–H bonds, regioselective α -methine alkynylation was achieved. In addition,

even in the presence of α -methine and linear- α -methylene C–H bonds, cyclic- α -methylene C–H bonds were regiospecifically alkynylated in this catalytic system. Considering the frequent appearance of α -methylene-substituted amines, especially cyclic methylene-substituted ones, in pharmaceutical fields, this catalytic system will be beneficial for synthesizing unexplored drug candidates and synthetic building blocks. Moreover, O₂ works as the sole oxidant in this reaction, which leads to environmentally-friendly alkynylation and no hydrogenation of products like the previous report of Beller and coworkers (*Chem. Commun.* **2010**, 46, 1956–1958).

Based on the aforementioned discussion, we summarized the previous main reports (including your introducing ones) on oxidative α -methylene C–H functionalizations of tertiary amines except for benzylic amines, *N*-protected amines, and symmetric amines and compared this study with the previous main reports in Supplementary Table 1. Obviously, this work is prominent in terms of demonstrated tertiary amine substrate types applicable to α -methylene selective functionalization with the number of the substrates and α -methylene selectivity vs α -methyl, vs α -methylene (cyclic or linear), and vs α -methine, showing the general α -methylene regiospecificity. During the revision of this paper, two important related reports were published (*Catal. Sci. Technol.* **2022**, 12, 1922–1933; *J. Am. Chem. Soc.* **2021**, 143, 18952–18959), and these reports are also summarized in Supplementary Table 1. Slowing et al. reported α -methylene-oxygenation reactions and a few limited examples on α -methylene alkynylation of tertiary amines via β -hydride elimination in the presence of an Au nanoparticle catalyst supported on mesoporous silica with pyridyl groups (*Catal. Sci. Technol.* **2022**, 12, 1922–1933), while the yields (8–16%) and turnover numbers (2–4) were quite low and insufficient for organic synthetic applications. Additionally, Schoenebeck and Rovis et al. mentioned the similar regioselectivity problem in the field of α -C–H functionalization of tertiary amines and realized regioselective oxidative α -C–H alkylation of tertiary amines at the more-substituted positions by utilizing the Curtin–Hammett principle via reversible and fast HAT catalysis (*J. Am. Chem. Soc.* **2021**, 143, 18952–18959); however, in principle, the unique regioselectivity is derived not from the amine oxidation step but from the reaction between amino alkyl radicals and electrophiles, which limits the functionalization types to reactions like Giese radical addition, and regioselectivity control between sterically hindered positions (e.g., linear methylene vs cyclic methylene) is quite difficult.

Thus, this report constitutes the first general example of an α -methylene-specific oxidative C–H alkynylation of tertiary amines via the regioselective amine oxidation step to form iminium cations, which brings about breakthrough of this field and paves the way for other various α -methylene-specific functionalizations.

“Supplementary Table 1. Comparison of this work with previous main reports on oxidative α -methylene C–H functionalisations of tertiary amines except for benzylic amines, *N*-protected amines, and symmetric amines.

Reference	Nucleophile or Reaction	Demonstrated tertiary amine substrate type applicable to α -methylene selective functionalization (number of the substrates)	α -Methylene selectivity			Reaction intermediate
			vs Methyl	vs Methylene	vs Methine	
this work	alkyne	linear, cyclic, unsymmetrical (17)	methylene	cyclic methylene	methylene	iminium cation
44	silylalkyne	cyclic (4)	cyclic methylene	cyclic methylene	—	iminium cation
45	alkyne	linear, cyclic (3) (low yields:8–16%)	cyclic methylene	—	methylene	iminium cation
51	iodoalkyne	cyclic (1)	methyl	cyclic methylene	—	amino alkyl radical
18	dicyanobenzene	cyclic (4)	—	cyclic methylene	methylene	amino alkyl radical
33	cyclization	cyclic (4) (minor α -methine oxidation)	—	cyclic methylene	methylene	iminium cation
46	aryl isocyanate	cyclic (3) (minor α -methyl oxidation)	cyclic methylene	—	methylene	amino alkyl radical
47	ketone	cyclic (1)	cyclic methylene	—	—	iminium cation
48	cyclization	cyclic, unsymmetrical (6)	methyl	cyclic methylene	methylene	iminium cation
49	bromoalkene	cyclic (4)	cyclic methylene	both	—	amino alkyl radical
50	bromobenzene	cyclic (1) (exceptional substrate)	both	—	methyl	amino alkyl radical
30	alkene	linear, cyclic, unsymmetrical (20) (Curtin–Hammet principle)	methylene	both	methine	amino alkyl radical
32	NaCN	linear, cyclic (12) (minor α - linear methylene oxidation)	both	cyclic methylene	methylene	iminium cation
33	NaCN	cyclic (3) (minor α -methyl oxidation)	cyclic methylene	cyclic methylene	—	iminium cation
36	NaCN	linear, cyclic, unsymmetrical (8) (minor α -methyl, linear methylene oxidation)	cyclic methylene	cyclic methylene	methylene	iminium cation
37	oxygenation	cyclic, unsymmetrical (3)	cyclic methylene	—	—	iminium cation
39	oxygenation	cyclic, unsymmetrical (20)	cyclic methylene	cyclic methylene	—	iminium cation
40	oxygenation	linear, cyclic (11)	methylene	cyclic methylene	—	iminium cation
45	oxygenation	linear, cyclic (7)	methylene	cyclic methylene	—	iminium cation

” (see SI, page 24)

18. McNally, A., Prier, C. K. & MacMillan, D. W. C. Discovery of an α -amino C–H arylation reaction using the strategy of accelerated serendipity. *Science* **334**, 1114–1117 (2011).
30. Shen, Y., Funez-Ardoiz, I., Schoenebeck, F. & Rovis T. Site-selective α -C–H functionalization of trialkylamines via reversible hydrogen atom transfer catalysis. *J. Am. Chem. Soc.* **143**, 18952–18959 (2021).
32. Chiba, T. & Takata, Y. Anodic cyanation of tertiary aliphatic and heterocyclic amines. *J. Org. Chem.* **42**, 2973–2977 (1977).
33. Chen, C.-K., Hortmann, A. G. & Marzabadi, M. R. ClO₂ oxidation of amines: synthetic utility and a biomimetic synthesis of elaeocarpidine. *J. Am. Chem. Soc.* **110**, 4829–4831 (1988).
36. Yilmaz, O., Oderinde, M. S. & Emmert, M. H. Photoredox-catalyzed C α –H cyanation of unactivated secondary and tertiary aliphatic amines: late-stage functionalization and mechanistic studies. *J. Org. Chem.* **83**, 11089–11100 (2018).
37. Moriarty, R. M., Vaid, R. K. & Duncan, M. P. Hypervalent iodine oxidation of amines using iodosobenzene: synthesis of nitriles, ketones and lactams. *Tetrahedron Lett.* **29**, 6913–6916 (1988).
39. Griffiths, R. J., Burley, G. A. & Talbot, E. P. A. Transition-metal-free amine oxidation: a chemoselective strategy for the late-stage formation of lactams. *Org. Lett.* **19**, 870–873 (2017).
40. Jin, X., Kataoka, K., Yatabe, T., Yamaguchi, K. & Mizuno, N. Supported gold nanoparticles for efficient α -oxygenation of secondary and tertiary amines into amides. *Angew. Chem. Int. Ed.* **55**, 7212–7217 (2016).
44. Jovel, I., Prateptongkum, S., Jackstell, R., Vogl, N., Weckbecker, C. & Beller, M. α -Functionalization of non-activated aliphatic amines: ruthenium-catalyzed alkynylations and alkylations. *Chem. Commun.* **46**, 1956–1958 (2010).
45. Chatterjee, P., Wang, H., Manzano, J. S., Kanbur, U., Sadow, A. D. & Slowing I. I. Surface ligands enhance the catalytic activity of supported Au nanoparticles for the aerobic α -oxidation of amines to amides. *Catal. Sci. Technol.* **12**, 1922–1933 (2022).
46. Yoshimitsu, T., Matsuda, K., Nagaoka, H., Tsukamoto, K. and Tanaka, T. Radical fixation of functionalized carbon resources: α -sp³C–H carbamoylation of tertiary amines with aryl isocyanates. *Org. Lett.* **24**, 5115–5118 (2007).
47. Sud, A., Sureshkumar, D. & Klussmann, M. Oxidative coupling of amines and ketones by combined vanadium- and organocatalysis. *Chem. Commun.* 3169–3171 (2009).
48. Deb, M. L., Dey, S. S., Bento, I., Barros, M. T. & Maycock, C. D. Copper-catalyzed regioselective intramolecular oxidative α -functionalization of tertiary amines: an efficient synthesis of dihydro-1,3-oxazines. *Angew. Chem. Int. Ed.* **52**, 9791–9795 (2013).
49. Sølvhøj, A., Ahlburg, A. & Madsen, R. Dimethylzinc-initiated radical coupling of β -bromostyrenes with ethers and amines. *Chem. Eur. J.* **21**, 16272–16279 (2015).
50. Ueno, R., Ikeda, Y. & Shirakawa, E. *tert*-Butoxy-radical-promoted α -arylation of alkylamines with aryl halides. *Eur. J. Org. Chem.* 4188–4193 (2017).
51. Ma, L., Shi, X., Li, X. & Shi, D. Iron-catalyzed alkynylation of tertiary aliphatic amines with 1-iodoalkynes to synthesize propargylamines. *Org. Chem. Front.* **5**, 3515–3519 (2018).

”(see pages 27–29, references 18, 30, 32, 33, 36, 37, 39, 40, 44–51)

Based on the aforementioned discussion, we revised the manuscript and SI as follows and as shown above.

“Similarly, the regioselectivity of the amine oxidation cannot be controlled in other reported mechanisms such as hydrogen atom transfer (HAT)/SET (Path B)^{24,25}, SET/HAT (Path C)^{26,27} and the Polonovski–Potier reaction via amine oxide formation (Path D)²⁸ (Supplementary Fig. 1), partly because α -C–H bonds of tertiary amines often possess nearly similar bond dissociation energies (e.g. 1-methylpiperidine: 92 kcal/mol for methyl vs 91 kcal/mol for methylene)^{29,30}. In particular, the substrate scope for oxidative α -methylene functionalisations is generally limited to *N*-substituted benzylic amines like tetrahydroisoquinolines (THIQs), *N*-protected amines and symmetric amines (Fig. 1b)^{7–9,14–20}. Considering the frequent appearance of α -methylene-substituted amines, especially cyclic methylene-substituted ones, in pharmaceutical fields^{2–4,31}, the development of an oxidative regioselective α -methylene C–H functionalisation is highly desirable.” (see page 4, lines 2–11)

“In addition, during the revision of this paper, Slowing et al. also reported α -methylene-oxygenation reactions and a few limited examples on α -methylene alkynylation of tertiary amines via β -hydride elimination in the presence of an Au nanoparticle catalyst supported on mesoporous silica with pyridyl groups, while the yields (8–16%) and turnover numbers (2–4) were quite low and insufficient for organic synthetic applications, and the detailed reaction mechanism for the amine oxidation was unrevealed⁴⁵. Quite recently, Schoenebeck and Rovis et al. realised regioselective oxidative α -C–H alkylation of tertiary amines at the more-substituted positions by utilising the Curtin–Hammett principle via reversible and fast HAT catalysis³⁰; however, in principle, the unique regioselectivity is derived not from the amine oxidation step but from the reaction between amino alkyl radicals and electrophiles, which limits the types of functionalisation to reactions like Giese radical addition, and regioselectivity control between sterically hindered positions (e.g., linear methylene vs cyclic methylene) is quite difficult³⁰. In addition to the aforementioned reports, although a few reports exist on oxidative functionalisation systems showing very limited examples of α -cyclic methylene functionalisations^{33,46–51}, a wide range of tertiary amines including unsymmetrical ones cannot be utilised in the oxidative α -methylene selective functionalisation reactions developed to date (the previous main reports on oxidative α -methylene C–H functionalisations of tertiary amines except for benzylic amines, *N*-protected amines and

symmetric amines are summarized in Supplementary Table 1). Considering this background, the development of novel general systems for the regioselective α -methylene functionalisation of tertiary amines containing a regioselective amine oxidation step would be an important breakthrough.

Herein, we have developed an unusual oxidative regiospecific α -methylene C–H functionalisation of tertiary amines via iminium cation formation with alkynes that produces various propargylic amines, which are widely used in organic synthesis and the pharmaceutical fields⁵², by utilising a combination of Zn salts and hydroxyapatite-supported Au nanoparticles (Au/HAP) as a catalytic system (Fig. 1d). The present catalytic system was applicable to a variety of aerobic α -methylene alkynylations of tertiary amines and afforded unreported propargylic amines, including cyclic derivatives except for THIQs and *N*-protected amines, which are difficult to synthesise using traditional methods^{53–59}. Surprisingly, even in the presence of α -methine and linear- α -methylene C–H bonds, cyclic- α -methylene C–H bonds were regiospecifically alkynylated in this catalytic system. Thorough experimental investigations revealed that the unusual α -methylene regiospecificity probably arises from a unique amine oxidation mechanism: a concerted one-proton/two-electron transfer from the amines to O₂ on the Au nanoparticle catalyst, which differs from the conventional catalytic amine oxidation mechanisms.” (see page 5, lines 9–18, page 6, lines 1–18, page 7, lines 1–6)

“

29. Wayner, D. D. M., Clark, K. B., Rauk, A., Yu, D. & Armstrong D. A. C-H bond dissociation energies of alkyl amines: radical structures and stabilization energies. *J. Am. Chem. Soc.* **119**, 8925–8932 (1997).
53. Li, Z. & Li, C.-J. CuBr-catalyzed efficient alkynylation of sp³ C-H bonds adjacent to a nitrogen atom. *J. Am. Chem. Soc.* **126**, 11810–11811 (2004).
54. Niu, M., Yin, Z., Fu, H., Jiang Y. & Zhao Y. Copper-catalyzed coupling of tertiary aliphatic amines with terminal alkynes to propargylamines via C-H activation. *J. Org. Chem.* **73**, 3961–3963 (2008).
55. Xu, X. & Li, X. Copper/diethyl azodicarboxylate mediated regioselective alkynylation of unactivated aliphatic tertiary methylamine with terminal alkyne. *Org. Lett.* **11**, 1027–1029 (2009).
56. Jin, X., Yamaguchi, K. & Mizuno, N. Aerobic cross-dehydrogenative coupling of terminal alkynes and tertiary amines by a combined catalyst of Zn²⁺ and OMS-2. *RSC Adv.* **4**, 34712–34715 (2014).
57. Teong, S. P., Yu, D., Sum Y. N. & Zhang Y. Copper catalysed alkynylation of tertiary amines with CaC₂ via sp³ C–H activation. *Green Chem.* **18**, 3499–3502 (2016).
58. Wang, G., Mao, Y. & Liu, L. Diastereoselectively complementary C–H functionalization enables access to structurally and stereochemically diverse 2,6-substituted piperidines. *Org. Lett.* **18**, 6476–6479 (2016).

59. Odachowski, M., Greaney, M. F. & Turner N. J. Concurrent biocatalytic oxidation and C–C bond formation via gold catalysis: one-pot alkynylation of *N*-alkyl tetrahydroisoquinolines. *ACS Catal.* **8**, 10032–10035 (2018).

” (see pages 27 and 29, references 29 and 53–59)

Comments

Second, the scope of the reaction is rather narrow with moderate efficiency. The work shares the routine scope of the above precedents. Whether the reaction can achieve excellent regio- and diastereoselectivities for unsymmetric tertiary amines? What about the asymmetric version of the transformation? Without these studies, the reviewer did not see the lightspots of the work to be published in *Nat. Commun.*

Responses

To achieve the broad substrate scope of tertiary amines including non-routine substrates and unsymmetric ones, we further optimized the reaction conditions by utilizing α -methine alkynylation of *N,N*-dimethylcyclohexylamine (**1m**) with **2a** as the model reaction. Considering that α -methine selective alkynylation was not demonstrated in the previous version, the achievement of the model reaction using **1m** is important. Under the previously optimized conditions, α -methine alkynylation of **1m** was difficult and afforded the desired propargylic amine (**3ma**) in low yield because of the preferential hydrolysis of the corresponding iminium cation compared with nucleophilic addition of **2a** (Supplementary Table 5, entry 1). When the amount of **2a** or ZnBr₂ was increased, the yield of **3ma** was improved due to the promotion of alkyne nucleophilic addition (Supplementary Table 5, entries 2–4). The addition of molecular sieves 4A (MS-4A) to remove H₂O in the reaction drastically increased the **3ma** yield (Supplementary Table 5, entries 5–8), and **3ma** was obtained in 84% GC yield under the optimized conditions using MS-4A (300 mg) (Supplementary Table 5, entry 7). Thus, α -methine selective alkynylation was successfully demonstrated without oxidation of its α -methyl group.

“Supplementary Table 5. Optimization of reaction conditions for α -methine alkylation of *N,N*-dimethylcyclohexylamine (1m**) with **2a**.^a**

Entry	2a [eq.]	ZnBr ₂ [mol%]	MS-4A [mg]	Conv. of 1m [%]	Yield [%]			
					3ma	4ma	5a	6a
1	2	20	0	89	12	1	3	19
2	4	20	0	77	16	1	1	11
3	6	20	0	87	23	1	1	7
4	6	40	0	99	35	<1	<1	10
5	6	20	100	72	28	1	<1	<1
6	6	20	200	84	52	1	<1	<1
7	6	20	300	97	84	<1	<1	<1
8	6	40	300	73	39	<1	1	<1

^aReaction conditions: **1m** (0.3 mmol), **2a** (0.6, 1.2, or 1.8 mmol), Au/HAP (100 mg, Au: 2.5 mol%), ZnBr₂ (20 or 40 mol%), MS-4A (0, 100, 200, or 300 mg), toluene (2 mL), 95 °C, O₂ (1 atm), 24 h. Conversions and yields were determined by gas chromatography analysis using biphenyl as internal standard. MS-4A = molecular sieves 4A.” (see SI, page 28)

The optimized conditions with MS-4A were also suitable for more difficult selective alkylation reactions of several amine substrates as shown in Table 3. When using 3-chloro-*N,N*-diethylpropan-1-amine (**1n**) as the substrate, surprisingly, the α -methylene alkylation product at the ethyl group (**3na**) was regioselectively obtained with the chloro group intact, suggesting that sterically non-hindered and/or electron-rich α -methylene C–H bonds can be selectively alkylation among linear- α -methylene positions. In fact, in some cases like the α -alkylation of **1a** and 1-ethylpiperidine (**1f**), the regioselectivity is correlated to the stability differences of the corresponding iminium cations; however, the selectivity of other cases, such as the reaction of 1-cyclohexylpiperidine (**1g**) or **1n**, is different from the stability order (Supplementary Fig. 31). These results suggested that this catalytic system is sensitive to kinetic factors like steric effects and electronic effects, which probably leads to the present unusual

regioselectivity. When using an unsymmetric amine, ethyl 1-methylpiperidine-2-carboxylate (**1o**), in spite of the presence of the cyclic- α -methine C–H bond, cyclic- α -methylene specific alkynylation occurred, and the corresponding *trans*-isomer was obtained stereoselectively (**3oa**) (*trans* : *cis* = 89 : 11). Likewise, the cyclic- α -methylene selective alkynylation of nicotine (**1p**), a kind of unsymmetric amines, proceeded *trans*-stereoselectively (**3pa**) (*trans* : *cis* = 81 : 19) with almost no α -methine alkynylated product. These results demonstrated excellent regioselectivities and diastereoselectivities for unsymmetric tertiary amines. Moreover, when using 4-hydroxymethyl-1-methylpiperidine as the substrate, the *trans*- α -methylene-alkynylated product was also stereoselectively obtained (**3da**). The *trans*-stereoselectivity in the case of several substituted cyclic tertiary amines is possibly derived from the attack of the alkynyl species from the opposite side of the catalyst surface to the iminium cation adsorbed on the Au nanoparticles. In addition, albeit in low yields, cyclic- α -methylene selective alkynylation of buflomedil and cloperastine successfully proceeded to afford unreported propargylic amines (**3qa** and **3ra**), demonstrating late-stage functionalization of medicines possessing tertiary amine structures. Furthermore, in the presence of D₂O (3 mmol, three equivalents to **1a**), selectively β -deuterated propargylic amine **3aa- β -d₂** (deuteration ratio: 76%) was successfully synthesized (Supplementary Fig. 28), which will be beneficial in the synthesis of deuterated medicines.

Overall, the broad substrate scope of tertiary amines including non-routine substrates and unsymmetric ones was successfully achieved in the catalytic system of Au/HAP and ZnBr₂ with the excellent α -methylene regioselectivity and stereoselectivity for the alkynylation.

Table 3. Tertiary amine substrate scope of the combined catalytic system comprising Au/HAP and Zn species.^a

^aReaction conditions: **1** (0.3 mmol), **2a** (0.6 mmol), Au/HAP (100 mg, Au: 2.5 mol%), ZnBr₂ (13 mg, 20 mol%), toluene (2 mL), 95 °C, O₂ (1 atm), 24 h. Isolated yields are shown. ^bAu/HAP (160 mg, 4 mol%). ^c**1** (1 mmol), **2a** (0.5 mmol), ZnBr₂ (11 mg, 10 mol%). ^dPhCF₃ (2 mL). ^eGC yield. ^fAu/HAP (200 mg). ^g**2a** (1.8 mmol), MS-4A (300 mg). The ratios of *cis/trans*-isomers were determined by ¹H NMR analysis or isolated yields." (see page 32, Table 3)

Supplementary Fig. 31 Comparison of regioselectivity to Au/HAP and ZnBr₂-catalyzed α -alkynylation of several selected tertiary amines (shown in Table 3) with Gibbs free energy difference given by DFT calculation between iminium cations derived from dehydrogenation at the cyclic methylene positions and the counterparts. Equilibrium constants (K) were calculated from $G^\circ = -RT \ln K$ ($T = 298.15 \text{ K}$). The number of α -C-H bonds of amines was taken into consideration on the shown ratios of iminium cations based on DFT calculation.” (see SI, page 55)

Supplementary Fig. 28 (a) β-Deuterated propargylic amine synthesis. Reaction conditions are indicated in this figure, and the yield was determined by isolation. The deuteration ratio was determined by ¹H NMR. (b) ¹H NMR, (c) ²H NMR, and (d) ¹³C NMR spectra of the isolated 3aa-β-d₂” (see SI, page 52)

Unfortunately, several amine substrates were not applicable to this alkylation as summarized in Supplementary Fig. 10. In addition to the previously demonstrated limitation, ethyl 3-(dimethylamino)propanoate, *N,N,N',N'*-tetramethylethylenediamine, 4-chloro-1-methylpiperidine, 1-methylpiperidin-4-one, (4-methylpiperazin-1-yl)(phenyl)methanone, and 4-(pyrrolidin-1-yl)benzaldehyde were inapplicable to this alkylation possibly because of the chelation, electron deficient α -methylene positions, aldehyde oxidation, *etc.* Moreover, when using amitriptyline or tolperison as the substrate, the complex mixture of compounds was observed. Although this catalytic system has the limitation of substrate scope, the applicability to 19 examples of structurally diverse tertiary amines and 15 examples of alkynes with various functional groups was successfully demonstrated. Considering the aforementioned excellent regioselectivity, this report constitutes a general example of an α -methylene-specific oxidative C–H alkylation of tertiary amines.

“

Supplementary Fig. 10 Limitation of amine substrate scope.^a Yields were determined by GC or ¹H NMR analysis. ^aReaction conditions: **1** (1 mmol), **2a** (0.5 mmol), Au/HAP (100 mg, Au: 1.5 mol%), ZnBr₂ (11 mg, 10 mol%), PhCF₃ (2 mL), 95 °C, O₂ (1 atm), 24 h, ^b20 h. ^cToluene (2 mL). ^d**1** (0.3 mmol), **2a** (0.6 mmol), Au/HAP (160 mg, 4 mol%), ZnBr₂ (13 mg, 20 mol%). ^e**2a** (1.8 mmol), MS-4A (300 mg).” (see SI, page 37)

We also tried the asymmetric α -methylene alkylation of tertiary amines based on this catalytic system. In general, to achieve asymmetric reactions, additional chiral ligands should be present in the reaction system. Although ZnBr_2 was used as the cocatalyst under the optimized conditions without ligands, $\text{Zn}(\text{OTf})_2$ was more suitable for the alkylation in the presence of a ligand like 2,2'-bipyridyl (bpy) than ZnBr_2 possibly because ligands induced the removal of Br^- from ZnBr_2 to inhibit the Au nanoparticle catalysis (Table A). Therefore, we used $\text{Zn}(\text{OTf})_2$ as the cocatalyst for the experiment of asymmetric alkylation reactions with chiral ligands. By referring to the previous reports (e.g., *Chem. Eur. J.* **2013**, *19*, 11992–11998; *Org. Lett.* **2010**, *12*, 5716–5719; *Asian J. Org. Chem.* **2018**, *7*, 1033–1053; *Tetrahedron Asymmetry* **2015**, *26*, 41–45), (*R*)-(+)-1,1'-bi-2-naphthol ((*R*)-BINOL), (*R*)-(+)-2,2'-bis(diphenylphosphino)-1,1'-binaphthyl ((*R*)-BINAP), (*R,R*)-2,6-bis(4-isopropyl-2-oxazolin-2-yl)pyridine ((*R,R*)-*i*Pr-PyBOX), and (+)-sparteine were utilized as the chiral ligand for the alkylation of **1a** with **2a** in the presence of Au/HAP and $\text{Zn}(\text{OTf})_2$ (Table B). At 95 °C, the optimized temperature for the alkylation of **3aa**, the enantiomeric ratios of **3aa** were almost 50/50 in the presence of any chiral ligands (Table B, entries 6–9). The yield of **3aa** was decreased by the presence of (*R*)-BINAP or (+)-sparteine while (*R*)-BINOL and (*R,R*)-*i*Pr-PyBOX did not largely affect the **3aa** yield (Table B, entries 1 and 6–9). To improve the enantiomeric ratio, the temperature effect on the **3aa** yield was investigated, and it was revealed that even at 70 °C the desired alkylation proceeded albeit in 23% yield for 24 h (Table B, entries 2–5). Then, we carried out the alkylation with chiral ligands at 70 °C for 48 h; however, the enantiomeric ratios were not improved in any cases (Table B, entries 10–13). In general, when using heterogeneous catalysts, it is quite difficult to achieve asymmetric reactions. We think that additional breakthrough is necessary for the achievement of α -methylene selective asymmetric alkylation of tertiary amines. Based on the aforementioned introduction, substrate scope, regioselectivity, and stereoselectivity of this study, we believe that the present work is sufficiently suitable for the publication in *Nature Communications* without achieving the asymmetric alkylation. We will continue the study of the asymmetric alkylation in the future.

Table A. The effect of Zn cocatalysts and 2,2'-bipyridyl on the yield of the α -methylene-selective alkylation of 1-methylpiperidine (**1a**) with phenylacetylene (**2a**).^a

Entry	Zn cocatalyst	bpy [mol%]	Yield [%]
			3aa
1	ZnBr ₂	0	65
2	ZnBr ₂	10	19
3	Zn(OTf) ₂	0	39
4	Zn(OTf) ₂	10	44

^aReaction conditions: **1a** (0.5 mmol), **2a** (0.5 mmol), Au/HAP (1.5 mol%), Zn cocatalyst (10 mol%), bpy (0 or 10 mol%), toluene (2 mL), 95 °C, O₂ (1 atm), 24 h. Yields were determined by gas chromatography analysis using biphenyl as an internal standard.

Table B. The effect of temperatures and chiral ligands on the yield and enantiomer ratio of the alkylation of 1-methylpiperidine (**1a**) with phenylacetylene (**2a**).^a

Entry	Temperature [°C]	Ligand	Yield [%]		Enantiomeric ratio
			3aa		
1	95	none	61	—	—
2	90	none	56	—	—
3	80	none	41	—	—
4	70	none	23	—	—
5	60	none	8	—	—
6	95	(R)-BINOL	60	51/49	—
7	95	(R)-BINAP	18	49/51	—
8	95	(R,R)- i Pr-PyBOX	66	46/54	—
9	95	(+)-sparteine	44	49/51	—
10 ^b	70	(R)-BINOL	33	51/49	—
11 ^b	70	(R)-BINAP	1	—	—
12 ^b	70	(R,R)- i Pr-PyBOX	40	47/53	—
13 ^b	70	(+)-sparteine	16	46/54	—

^aReaction conditions: **1a** (1 mmol), **2a** (0.5 mmol), Au/HAP (1.5 mol%), Zn(OTf)₂ (10 mol%), ligand (0 or 10 mol%), PhCF₃ (2 mL), O₂ (1 atm), 24 h. ^b48 h. Yields were determined by gas chromatography analysis using biphenyl as an internal standard. Enantiomeric ratios were determined by HPLC analysis using Daicel chiralcel OD-H column. —: The enantiomeric ratio was not determined due to the absence of chiral ligands or too low **3aa** yield.

Based on the aforementioned results and discussion, we revised the manuscript and SI as follows and as shown above.

“On the other hand, α -methine alkylation of *N,N*-dimethylcyclohexylamine (**1m**) was difficult because of the preferential hydrolysis of the corresponding iminium cation compared with nucleophilic addition of **2a** (Supplementary Table 5, entry 1). When the amount of **2a** or ZnBr₂ was increased, the yield of the desired propargylic amine (**3ma**) was improved due to the promotion of alkyne nucleophilic addition (Supplementary Table 5, entries 2–4). The addition of molecular sieves 4A (MS-4A) to remove H₂O in the reaction drastically increased the **3ma** yield (Supplementary Table 5, entries 5–8), and **3ma** was obtained in 84% GC yield (64% isolated yield) under the optimized conditions using MS-4A (300 mg) (Table 3, **3ma**, Supplementary Table 5, entry 7). The conditions with MS-4A were also suitable for more difficult selective alkylation reactions of the following several amine substrates. When using 3-chloro-*N,N*-diethylpropan-1-amine as the substrate, surprisingly, the α -methylene alkylation product at the ethyl group (**3na**) was regiospecifically obtained with the chloro group intact, suggesting that sterically non-hindered and/or electron-rich α -methylene C–H bonds can be selectively alkylated among linear- α -methylene positions. Even in the presence of a cyclic- α -methine C–H bond with an ester group, cyclic- α -methylene selective alkylation occurred to produce the corresponding *trans*-isomer stereoselectively (**3oa**). Likewise, the cyclic- α -methylene selective alkylation of nicotine proceeded *trans*-stereoselectively (**3pa**), and as mentioned above, **3da** was also obtained as the *trans*-isomer stereoselectively. The *trans*-stereoselectivity is possibly derived from the attack of the alkynyl species from the opposite side of the catalyst surface to the iminium cation adsorbed on the Au nanoparticles. In addition, albeit in low yields, cyclic- α -methylene selective alkylation of bufomedil and cloperastine successfully proceeded to afford unreported propargylic amines (**3qa** and **3ra**), demonstrating late-stage functionalisation of medicines possessing tertiary amine structures. Unfortunately, several amine substrates were not applicable to this alkylation as summarized in Supplementary Fig. 10. For example, as expected from the high methylene selectivity, *N,N*-dimethylaniline was not converted to the desired product. The inability of Au nanoparticles to catalyze amide oxidations⁴⁰ resulted in no reaction when using an amine protected by *tert*-butoxycarbonyl group. This reaction system was not applicable to the α -alkylation of either quinuclidine or a secondary amine like piperidine. In the former case, the formation of the corresponding iminium cation would be forbidden by Bredt’s rule, which supports the expected reaction pathway involving the production of iminium cation intermediates.” (see page 12, lines 16–18, page 13, lines 1–18, page 14, lines 1–9)

“In fact, in the presence of a large amount of D₂O (3 mmol, three equivalents to **1a**), selectively β-deuterated propargylic amine **3aa-β-d₂** (deuteration ratio: 76%) was successfully synthesised in 44% yield (Supplementary Fig. 28), which will be beneficial in the synthesis of deuterated medicines⁷⁰.” (see page 18, lines 17–18, page 19, lines 1 and 2)

“Thus, in some cases like the α-alkynylation of **1a** and 1-ethylpiperidine (**1f**), the regioselectivity is correlated to the stability differences of the corresponding iminium cations; however, the selectivity of other cases, such as the reaction of 1-cyclohexylpiperidine (**1g**) or 3-chloro-*N,N*-diethylpropan-1-amine (**1n**), is different from the stability order because of kinetic factors like steric effects and electronic effects (Supplementary Fig. 31), which probably leads to the unusual α-methylene selectivity.” (see page 19, lines 14–17, page 20, lines 1 and 2)

“In the presence of α-methine and linear-α-methylene C–H bonds, cyclic-α-methylene C–H bonds were regiospecifically alkynylated, and *trans*-stereoselective alkynylation was also observed in the case of substituted cyclic tertiary amines. Most of the as-produced propargylic amines exhibit unprecedented structures including alkynylated bufloxedil and cloperastine, demonstrating the great utility of this novel transformation to synthesise unexplored drug candidates and synthetic building blocks.” (see page 23, lines 4–9)

“70. Pirali, T., Serafini, M., Cargnin, S. & Genazzani, A. A. Applications of deuterium in medicinal chemistry. *J. Med. Chem.* **62**, 5276–5297 (2019).” (see page 29, reference 70)

“

3ma (CAS No. 1313814-49-1)

N,N-dimethyl-1-(phenylethynyl)cyclohexan-1-amine: 64% isolated yield (eluent: hexane/EtOAc = 4/6). ¹H NMR (500 MHz, CDCl₃, TMS): δ 1.19–1.27 (m, 1H), 1.44–1.49 (m, 2H), 1.59–1.75 (m, 5H), 2.05–2.11 (m, 2H), 2.36 (s, 6H), 7.27–7.32 (m, 3H), 7.42–7.47 (m, 2H). ¹³C-¹H NMR (125 MHz, CDCl₃, TMS): δ 23.2, 25.7, 36.5, 39.5, 59.9, 86.5, 89.5, 123.8, 127.9, 128.3, 131.9. MS (70 eV, EI): *m/z* (%): 227 (20) [*M*⁺], 212 (31), 198 (11), 185 (16), 184 (100), 170 (17), 150 (13), 142 (10), 141 (30), 128 (11), 115 (24). Anal. Calcd. for C₁₆H₂₁N·0.1H₂O: C, 83.86; H, 9.32; N, 6.11. Found: C, 83.86; H, 8.88; N, 5.65.

3na

***N*-(3-chloropropyl)-*N*-ethyl-4-phenylbut-3-yn-2-amine**: 28% isolated yield (eluent: hexane/EtOAc = 9/1). ^1H NMR (500 MHz, CDCl_3 , TMS): δ 1.10 (t, $J = 7.2$ Hz, 3H), 1.40 (d, $J = 7.1$ Hz, 3H), 1.87–2.00 (m, 2H), 2.49–2.60 (m, 2H), 2.64–2.71 (m, 1H), 2.76–2.84 (m, 1H), 3.59–3.68 (m, 2H), 3.90 (q, $J = 7.1$ Hz, 1H), 7.27–7.31 (m, 3H), 7.39–7.44 (m, 2H). ^{13}C - $\{^1\text{H}\}$ NMR (125 MHz, CDCl_3 , TMS): δ 14.0, 20.5, 31.7, 43.5, 45.7, 47.6, 48.7, 84.2, 89.5, 123.6, 128.0, 128.4, 131.8. MS (70 eV, EI): m/z (%): 249 (3) [M^+], 237 (5), 236 (34), 235 (17), 234 (100), 186 (17), 130 (11), 129 (78), 128 (36), 127 (15), 115 (16), 103 (5), 77 (8), 58 (11), 56 (5). Anal. Calcd. for $\text{C}_{15}\text{H}_{20}\text{ClN}$: C, 72.13; H, 8.07; N, 5.61. Found: C, 72.62; H, 7.82; N, 5.33.

3oa

***trans*-ethyl 1-methyl-6-(phenylethynyl)piperidine-2-carboxylate, *cis*-ethyl 1-methyl-6-(phenylethynyl)piperidine-2-carboxylate**: *trans*-isomer: 39% isolated yield (eluent: hexane/EtOAc = 6/4). ^1H NMR (500 MHz, CDCl_3 , TMS): δ 1.29 (t, $J = 7.1$ Hz, 3H), 1.60–1.70 (m, 2H), 1.77–1.91 (m, 3H), 1.94–2.01 (m, 1H), 2.40 (s, 3H), 3.26 (dd, $J = 10.6$ and 3.0 Hz, 1H), 4.01 (t, $J = 3.6$ Hz, 1H), 4.22 (q, $J = 7.1$ Hz, 2H), 7.29–7.33 (m, 3H), 7.43–7.47 (m, 2H). ^{13}C - $\{^1\text{H}\}$ NMR (125 MHz, CDCl_3 , TMS): δ 14.4, 19.3, 30.0, 30.7, 41.8, 53.8, 60.9, 63.1, 86.1, 87.5, 123.2, 128.2, 128.4, 131.9, 173.8. MS (70 eV, EI): m/z (%): 271 (2) [M^+], 199 (15), 198 (100), 170 (6), 169 (6), 167 (10), 142 (8), 141 (38), 128 (8), 115 (19), 102 (5), 96 (45), 91 (6), 85 (9). Anal. Calcd. for $\text{C}_{17}\text{H}_{21}\text{NO}_2$: C, 75.25; H, 7.80; N, 5.16. Found: C, 75.52; H, 7.45; N, 4.93. *cis*-isomer: 5% isolated yield (eluent: hexane/EtOAc = 6/4). ^1H NMR (500 MHz, CDCl_3 , TMS): δ 1.29 (t, $J = 7.1$ Hz, 3H), 1.35–1.47 (m, 1H), 1.70–1.80 (m, 1H), 1.84–1.94 (m, 3H), 2.00–2.08 (m, 1H), 2.50 (s, 3H), 2.77 (dd, $J = 11.5$ and 2.7 Hz, 1H), 3.02 (dd, $J = 11.1$ and 3.1 Hz, 1H), 4.24 (qd, $J = 7.1$ and 1.0 Hz, 2H), 7.27–7.30 (m, 3H), 7.39–7.42 (m, 2H). ^{13}C - $\{^1\text{H}\}$ NMR (125 MHz, CDCl_3 , TMS): δ 14.4, 23.5, 29.8, 32.7, 42.5, 56.9, 61.0, 69.0, 84.4, 89.4, 123.4, 128.2, 128.3, 131.7, 173.2. MS (70 eV, EI): m/z (%): 271 (1) [M^+], 199 (16), 198 (100), 167 (9), 142 (7), 141 (36), 128 (7), 115 (17), 96 (35), 85 (6).

3pa

***trans*-3-(1-methyl-5-(phenylethynyl)pyrrolidin-2-yl)pyridine, *cis*-3-(1-methyl-5-(phenylethynyl)pyrrolidin-2-yl)pyridine, 3-(1-methyl-2-(phenylethynyl)pyrrolidin-2-yl)pyridine:** 32% isolated yield in total (eluent: hexane/EtOAc = 4/6) (the ratio determined by ^1H NMR: *trans*-isomer/*cis*-isomer/*regio*-isomer = 77/18/5). Anal. Calcd. for $\text{C}_{18}\text{H}_{18}\text{N}_2 \cdot 0.1\text{H}_2\text{O}$: C, 81.84; H, 6.94; N, 10.60. Found: C, 81.57; H, 6.80; N, 10.50.

trans-isomer: ^1H NMR (500 MHz, CDCl_3 , TMS): δ 1.71–1.79 (m, 1H), 2.07 (dddd, $J = 12.3, 9.2, 3.4$ and 1.2 Hz, 1H), 2.29 (s, 3H), 2.31–2.39 (m, 1H), 2.43–2.53 (m, 1H), 3.63 (dd, $J = 8.8$ and 7.1 Hz, 1H), 4.27 (d, $J = 6.5$ Hz, 1H), 7.25–7.36 (m, 4H), 7.44–7.49 (m, 2H), 7.70 (dt, $J = 7.8$ and 1.8 Hz, 1H), 8.51 (dd, $J = 4.8$ and 1.7 Hz, 1H), 8.57 (d, $J = 1.9$ Hz, 1H). ^{13}C - $\{^1\text{H}\}$ NMR (125 MHz, CDCl_3 , TMS): δ 31.0, 33.9, 36.5, 57.0, 64.9, 86.5, 87.5, 123.3, 123.7, 128.1, 128.4, 131.9, 135.1, 139.3, 148.8, 149.7. MS (70 eV, EI): m/z (%): 263 (10), 262 (60) [M^+], 261 (100), 234 (27), 233 (42), 219 (12), 193 (12), 185 (27), 184 (37), 156 (13), 155 (23), 142 (26), 129 (32), 128 (16), 119 (10), 118 (18), 115 (21).

cis-isomer: ^1H NMR (500 MHz, CDCl_3 , TMS): δ 1.77–1.87 (m, 1H), 2.10–2.29 (m, 3H), 2.30 (s, 3H), 3.22–3.27 (m, 1H), 3.31–3.37 (m, 1H), 7.26–7.34 (m, 4H), 7.45–7.49 (m, 2H), 7.79 (dt, $J = 7.8$ and 2.0 Hz, 1H), 8.52 (dd, $J = 4.8$ and 1.7 Hz, 1H), 8.56 (d, $J = 1.9$ Hz, 1H). ^{13}C - $\{^1\text{H}\}$ NMR (125 MHz, CDCl_3 , TMS): δ 30.9, 33.4, 39.2, 58.7, 68.7, 83.8, 89.6, 123.3, 123.8, 128.2, 128.4, 131.9, 135.2, 138.4, 149.1, 149.7. MS (70 eV, EI): m/z (%): 263 (11), 262 (62) [M^+], 261 (100), 234 (24), 233 (37), 219 (11), 193 (11), 185 (30), 184 (52), 156 (16), 155 (34), 143 (10), 142 (22), 129 (36), 128 (18), 127 (10), 118 (22), 115 (24).

regio-isomer: ^1H NMR (500 MHz, CDCl_3 , TMS): δ 1.96–2.53 (m, 4H), 2.20 (s, 3H), 2.72–2.78 (m, 1H), 3.22–3.27 (m, 1H), 7.25–7.36 (m, 4H), 7.44–7.54 (m, 2H), 8.02 (dt, $J = 8.0$ and 2.0 Hz, 1H), 8.49–8.59 (m, 1H), 9.00 (d, $J = 1.7$ Hz, 1H). ^{13}C - $\{^1\text{H}\}$ NMR (125 MHz, CDCl_3 , TMS): δ 21.7, 36.1, 44.8, 53.8, 67.3, 86.6, 89.1, 123.1, 123.6, 128.2, 128.3, 131.9, 134.6, 138.6, 148.7, 149.2. MS (70 eV, EI): m/z (%): 262 (12) [M^+], 261 (9), 233 (10), 185 (18), 184 (100), 115 (6).

3qa

4-(2-(phenylethynyl)pyrrolidin-1-yl)-1-(2,4,6-trimethoxyphenyl)butan-1-one: 12% isolated yield (eluent: EtOAc). $^1\text{H NMR}$ (500 MHz, CDCl_3 , TMS): δ 1.75–2.04 (m, 5H), 2.12–2.23 (m, 1H), 2.43–2.55 (m, 2H), 2.73–2.91 (m, 4H), 3.54–3.61 (m, 1H), 3.74 (s, 6H), 3.80 (s, 3H), 6.07 (s, 2H), 7.26–7.30 (m, 3H), 7.39–7.43 (m, 2H). $^{13}\text{C}-\{^1\text{H}\}$ NMR (125 MHz, CDCl_3 , TMS): δ 22.3, 23.2, 32.0, 43.2, 51.9, 53.2, 55.3, 55.5, 55.9, 84.4, 89.2, 90.7, 113.8, 123.6, 128.0, 128.3, 131.8, 158.2, 162.2, 204.6. MS (70 eV, EI): m/z (%): 407 (3) [M^+], 376 (16), 213 (17), 212 (100), 210 (12), 198 (23), 197 (40), 196 (73), 195 (75), 170 (34), 169 (25), 156 (10), 153 (10), 152 (12), 137 (11), 129 (10), 128 (22), 120 (17), 115 (17), 102 (12). Anal. Calcd. for $\text{C}_{25}\text{H}_{29}\text{NO}_4$: C, 73.69; H, 7.17; N, 3.44. Found: C, 73.33; H, 6.68; N, 3.21.

3ra

1-(2-((4-chlorophenyl)(phenyl)methoxy)ethyl)-2-(phenylethynyl)piperidine: 17% isolated yield (eluent: hexane/EtOAc = 8/2 (1st), toluene/EtOAc = 9/1 (2nd)). $^1\text{H NMR}$ (500 MHz, CDCl_3 , TMS): δ 1.50–1.76 (m, 4H), 1.81–1.88 (m, 2H), 2.54–2.70 (m, 2H), 2.79–2.89 (m, 2H), 3.56–3.66 (m, 2H), 3.82–3.86 (m, 1H), 5.37 (s, 1H), 7.22–7.35 (m, 12H), 7.41–7.46 (m, 2H). $^{13}\text{C}-\{^1\text{H}\}$ NMR (125 MHz, CDCl_3 , TMS): δ 20.7, 25.8, 31.5, 49.8, 52.8, 55.8, 67.3, 83.1, 86.4, 87.4, 123.5, 127.0, 127.6, 127.9, 128.2, 128.4, 128.45, 128.47, 131.7, 133.1, 141.0, 141.9. MS (70 eV, EI): m/z (%): 431 (7), 430 (6), 429 (19) [M^+], 229 (17), 228 (100), 212 (13), 210 (6), 203 (12), 202 (6), 201 (37), 200 (7), 185 (8), 184 (7), 182 (8), 170 (6), 169 (7), 168 (9), 167 (17), 166 (19), 165 (20), 122 (16), 91 (23), 79 (5). Anal. Calcd. for $\text{C}_{28}\text{H}_{28}\text{ClNO}$: C, 78.21; H, 6.56; N, 3.26. Found: C, 78.20; H, 6.46; N, 3.27.” (see SI, pages 19–22)

Eluent: hexane/EtOAc = 4/6 (1st), hexane/EtOAc = 3/7 (2nd).

Eluent: hexane/EtOAc = 4/6 (1st), hexane/EtOAc = 3/7 (2nd).

Eluent: hexane/EtOAc = 4/6 (1st), hexane/EtOAc = 3/7 (2nd). Peaks of the *regio*-isomer were picked up from the mixture of *trans/cis/regio*-isomers.

$^1\text{H}-^1\text{H}$ COSY of 3qa

”(see SI, pages 86–95)

Comments

Third, while a thorough experimental investigation was conducted, the hydride transfer mechanism for oxidative C-H functionalization is known, which should be cited (ex. *Org. Lett.* 2016, 18, 6476–6479; *Eur. J. Org. Chem.* 2010, 4460–4467; *J. Org. Chem.* 2015, 80, 16, 8150–8167).

Responses

Thank you for introducing several reports on the hydride transfer mechanism for oxidative C–H functionalization of tertiary amines.

First of all, under the revision of this article, we noticed that the concerted hydride transfer from tertiary amines to O₂ on Au nanoparticles could proceed without formation of hydride, i.e., proton-coupled electron transfer could be also proposed. Thus, we changed the expression from concerted hydride transfer to concerted one-proton/two-electron transfer in the revised manuscript and SI.

Although the introduced reports exhibit one-proton/two-electron (hydride) transfer from tertiary amines to oxidants, the substrate scopes are limited to *N*-protected amines without demonstrating α -methylene selectivity. In addition, (super)stoichiometric amounts of oxidants were indispensable for these previously reported reactions. By contrast, the present catalytic regiospecific α -methylene alkylation using Au/HAP and ZnBr₂ is applicable to a variety of tertiary amines as shown in Supplementary Table 1 and Table 3 and utilizes O₂ as the sole oxidant. To the best of our knowledge, a concerted one-proton/two-electron transfer from tertiary amines to O₂ via a catalyst exhibiting α -methylene selectivity is hitherto unknown.

Therefore, we added the introduced references to the revised manuscript with the additional sentences to clarify the novelty of the present amine oxidation mechanism as follows.

“Although one-proton/two-electron (hydride) transfer from tertiary amines to stoichiometric oxidants has been reported to date^{58,66,67}, to our knowledge, a concerted one-proton/two-electron transfer from tertiary amines to O₂ via a catalyst exhibiting α -methylene selectivity is hitherto unknown.” (see page 17, lines 15–18)

“58. Wang, G., Mao, Y. & Liu, L. Diastereoselectively complementary C–H functionalization enables access to structurally and stereochemically diverse 2,6-substituted piperidines. *Org. Lett.* **18**, 6476–6479 (2016).

66. Richter, H. & Mancheño, O. G. Dehydrogenative functionalization of C(sp³)–H bonds adjacent to a heteroatom mediated by oxoammonium salts. *Eur. J. Org. Chem.* 4460–4467 (2010).

67. Hamlin, T. A., Kelly, C. B., Ovian, J. M., Wiles, R. J., Tilley, L. J. & Leadbeater, N. E. Toward a unified mechanism for oxoammonium salt-mediated oxidation reactions: a theoretical and experimental study using a hydride transfer model. *J. Org. Chem.* **80**, 8150–8167 (2015).” (see page 29, references 58, 66, and 67)

Comments

Other comments:

1) The substrate scope is mainly focused on cyclic tertiary amines. More acyclic substrates having different kind of functional groups may be tested to show the substrate scope of the reaction further.

Responses

As shown above in our responses to your second comments (Table 3 and Supplementary Fig. 10), more acyclic substrates having different kind of functional groups such as *N,N*-dimethylcyclohexylamine (**1m**), 3-chloro-*N,N*-diethylpropan-1-amine (**1n**), ethyl 3-(dimethylamino)propanoate, *N,N,N',N'*-tetramethylethylenediamine, and amitriptyline, were tested for the present oxidative alkynylation, and **1m** and **1n** were applicable to this catalytic system under the optimized conditions using MS-4A (300 mg). In the case of **1m**, α -methine selective alkynylation was successfully demonstrated without oxidation of its α -methyl group. When using **1n** as the substrate, surprisingly, the α -methylene alkynylated product at the ethyl group (**3na**) was regiospecifically obtained with the chloro group intact, suggesting that sterically non-hindered and/or electron-rich α -methylene C–H bonds can be selectively alkynylated among linear- α -methylene positions. Thus, the substrate scope expansion to **1m** and **1n** elevated the utility of this catalytic oxidative alkynylation.

Comments

2) 3ka and 3la were obtained in 54% and 27% isolated yield. Are there any demethylated or dealkylated products generated?

Responses

First of all, in the present catalytic system using Au/HAP and ZnBr₂, the oxidation of propargylic amines was not observed in any cases. Moreover, secondary amines are not applicable to this oxidative alkynylation, and the oxidation of *N*-methyl groups did not proceed in this system as shown in Table 3 and Supplementary Fig. 10. Then, there are no demethylated or dealkylated products of **3ka** or **3la** in these cases. Meanwhile, in the case of **3ka** synthesis, a trace amount of dimethylamine, which can be

produced via oxidation at the α -methylene position of *N,N*-dimethyloctylamine (**1k**) followed by hydrolysis, was detected by GC-MS. Likewise, in the case of **3la** synthesis, diethylamine was generated in 13% yield via oxidation/hydrolysis of triethylamine (**1l**). The dealkylation reactions of amine substrates may be one possible reason for the moderate yields of **3ka** and **3la**.

Comments

3) Check the structure of compounds 3qa.

Responses

According to your comment, we tried to confirm the absolute configuration of the major diastereomer of **3qa** (**3sa**: revised version) by X-ray crystallography; however, the single crystal could not be obtained. On the basis of the aforementioned discussion on the *trans*-stereoselectivity, the diastereoselectivity to **3sa** was possibly derived from the alkyne nucleophilic addition to the iminium cation on the Au nanoparticles. Considering the steric effect on Au nanoparticles and the corresponding iminium cation structure after the optimization by DFT calculations (M06/6-31G(d,p)) as shown below (Figure A), the assumed major diastereomer of **3sa** might be (*R*^{*})-*N,N*-dimethyl-3-phenyl-1-(*trans*-2-phenylcyclopropyl)prop-2-yn-1-amine. However, based on the nucleophilic addition from Bürgi-Dunitz trajectory and the steric effect of phenylcyclopropyl group, the assumed major diastereomer of **3sa** might be (*S*^{*})-*N,N*-dimethyl-3-phenyl-1-(*trans*-2-phenylcyclopropyl)prop-2-yn-1-amine (the nucleophilic addition from the *Si*-face of the iminium cation seems to be difficult). Thus, it is difficult to determine the major diastereomer structure of **3sa** from the alkylation mechanism.

Figure A. The corresponding iminium cation structure of **1s** after the optimization by DFT calculations (M06/6-31G(d,p)).

On the other hand, this article mainly focuses on the unusual regioselectivity to the alkylation of tertiary amines, and many reports on diastereoselective reactions have

been published without determining the absolute configurations of all the reported products. In addition, **3sa** production is important for determining that hydrogen atom transfer does not occur in this amine oxidation by the confirmation of no ring-opened product formation, and the result is shown not in Table 3 about amine substrate scope but in Fig.2 about mechanistic studies. Therefore, we do not think that the determination of absolute configuration of the major diastereomer of **3sa** is necessary for the publication of this article.

Based on the aforementioned discussion, we added the following sentence to the revised manuscript.

“The diastereoselectivity is possibly due to the alkyne nucleophilic addition to the iminium cation on the Au nanoparticles as with the aforementioned *trans*-stereoselectivity in the case of **3da**, **3oa** and **3pa** or due to the steric effect of phenylcyclopropyl group.” (see page 15, lines 6–8)

Comments

4) SI: Please report the yields of the final products and insert them before the description of the NMR spectra.

Responses

According to your comments, we inserted the yields of the alkynylated products with eluents for column chromatography before the description of the NMR spectra in the revised SI. An example of **3aa** is shown as follows.

3aa (CAS No. 51498-55-6)

1-methyl-2-(phenylethynyl)piperidine (3aa): 78% isolated yield (eluent: hexane/EtOAc = 6/4). ¹H NMR (500 MHz, CDCl₃, TMS): δ 1.47–1.52 (m, 1H), 1.55–1.75 (m, 3H), 1.81–1.93 (m, 2H), 2.35–2.41 (m, 1H), 2.41 (s, 3H), 2.62–2.68 (m, 1H), 3.55 (brs, 1H), 7.27–7.32 (m, 3H), 7.42–7.46 (m, 2H). ¹³C–{¹H} NMR (125 MHz, CDCl₃, TMS): δ 20.9, 25.9, 31.9, 44.6, 52.1, 54.8, 86.2, 87.6, 123.5, 128.0, 128.3, 131.8. MS (70 eV, EI): *m/z* (%): 199 (44) [*M*⁺], 198 (54), 184 (11), 171 (23), 170 (100), 157 (22), 156 (16), 143 (11), 142 (36), 128 (28), 127 (14), 122 (25), 116 (15), 115 (44), 102 (13), 94 (11), 79 (10). Anal. Calcd. for C₁₄H₁₇N·0.25H₂O: C, 82.57; H, 8.66; N, 6.87. Found: C, 82.97; H, 8.49; N, 6.85.” (see SI, page 9)

Comments

5) In the ^{13}C NMR spectrum of **3ao**, better signal: noise is needed to properly identify the quartet with the largest splitting.

Responses

We isolated **3ao** again and obtained the ^{13}C NMR spectrum of **3ao** possessing the better signal/noise ratio. As a result, it was revealed that the attribution of the peak about its $\text{C}\equiv\text{C}$ bond was wrong in the previous version. The correct ^{13}C NMR spectral data and the ^{13}C NMR spectrum of **3ao** possessing the better signal/noise ratio were shown in the revised SI as follows.

“

3ao

1-methyl-2-(oct-1-yn-1-yl)piperidine (3ao): 26% isolated yield (eluent: hexane/EtOAc = 6/4). ^1H NMR (500 MHz, CDCl_3 , TMS): δ 0.89 (t, $J = 7.1$ Hz, 3H), 1.27–1.35 (m, 4H), 1.38–1.44 (m, 3H), 1.48–1.72 (m, 6H), 1.77–1.83 (m, 1H), 2.22 (td, $J = 7.0$ and 1.9 Hz, 2H), 2.32 (s, 3H), 2.28–2.36 (m, 1H), 2.57–2.60 (m, 1H), 2.92 (brs, 1H). ^{13}C - $\{^1\text{H}\}$ NMR (125 MHz, CDCl_3 , TMS): δ 14.2, 18.8, 21.0, 22.7, 25.9, 28.6, 29.2, 31.4, 32.4, 44.4, 52.2, 54.5, 77.9, 86.0. MS (70 eV, EI): m/z (%): 207 (26) [M^+], 206 (48), 192 (19), 178 (21), 164 (19), 151 (12), 150 (100), 137 (69), 136 (49), 134 (14), 124 (18), 123 (12), 122 (84), 120 (10), 110 (14), 109 (24), 108 (57), 107 (13), 98 (24), 96 (38), 95 (22), 94 (62), 93 (16), 91 (18), 84 (11), 82 (27), 81 (20), 80 (19), 79 (26), 77 (19), 70 (25), 68 (13), 67 (20), 65 (14), 57 (14), 55 (23), 53 (15). Anal. Calcd. for $\text{C}_{14}\text{H}_{25}\text{N}\cdot 0.25\text{H}_2\text{O}$: C, 79.37; H, 12.13; N, 6.61. Found: C, 79.28; H, 11.92; N, 6.34.” (see SI, page 14)

“

(see SI, page 73)

<To Reviewer # 2>

Comments

This manuscript is a sound contribution in the fields of Au nanoparticle catalysis and synthetic organic methodology as well. It is initiated by previous studies from the same group regarding the selective aerobic oxidation of amines into amides (ref 37), and herein it is shown that suitable acetylides can C-C couple on the thermodynamically most stable iminium cation, which has very few preceding examples in the literature. What is obscure and needs further attention is the mechanistic part.

Responses

Thank you for your kind review. According to your valuable and helpful comments, we did various additional experiments and revised the manuscript and the Supplementary Information (SI) thoroughly, especially about the introduction, substrate scopes, and mechanistic studies. We believe that the revised version is much improved and sufficiently suitable for the publication in *Nature Communications*. Please confirm the following responses, the revised manuscript, and the revised SI.

Comments

a) The possibility of the generation of an iminium cation as an intermediate appears reasonable, as it fits nicely with the regioselectivity of the C-C coupling, except the case of 3ga where alkynylation is kinetically driven. Thus, in the case of N-methyl piperidine as an example, one would expect the formation of the most stable endocyclic iminium cation versus the exocyclic (Supplementary Fig. 23), as occurs. The authors have studied amines 1q and 1r as probes to exclude the formation of radical cations, as no ring opening rearrangement is observed in their alkynylations. However, I would anticipate the iminium cation from 1q to undergo ring opening, as a-cyclopropyl probes are sensitive not only to radical but to carbocations as well. My impression is that an iminium cation bound on Au nanoparticle is the most possible intermediate. Due to the low polarity of Au-C bonds (JACS Au, 2021, 1, 362) the intermediate might not be exactly ionic but rather having partial charges that do not possible allow ring opening rearrangement in the case of 1q. There is an additional example of none phenylcyclopropyl ring opening (Org. Lett. 2019, 21, 5552) presumably for the same reasons. I believe that the mechanistic analysis needs revision, and furthermore, theoretical calculations on a model nanoparticle may help to clarify the situation.

Responses

Thank you for your important remarks. As you pointed out, it is well known that cyclopropylcarbinyl cations tend to undergo ring opening reactions easily. However, in the case of the radical clock for hydrogen atom transfer (HAT) (**1q**, revised version: **1s**), DFT scan calculation using Gaussian (M06/6-31G(d,p)) about the ring-opening reaction of the corresponding iminium cation of **1s** suggested that the ring-opened carbocation is quite unstable compared with the initial iminium cation (Supplementary Fig. 13b), while the similar DFT calculation suggested that the ring-opening reaction of the corresponding carbon-centered radical species proceeds quite easily (estimated activation energy: 6.8 kcal/mol) to form the stable ring-opened product (estimated energy difference between the ring-closed radical species and the ring-opened radical species: 7.2 kcal/mol) (Supplementary Fig. 13a). Thus, **1s** is assumed to function suitably as a radical clock without the ring opening from the corresponding iminium cation.

“(a)

(b)

Supplementary Fig. 13 DFT scan calculation using Gaussian about ring-opening reactions of the corresponding (a) carbon-centered radical species and (b) iminium cation species derived from **1s**, respectively.” (see SI, page 39)

In addition, as you mentioned, adsorption of the corresponding iminium cation of **1s** on the Au nanoparticles can be one reason for the no ring opening of **1s**. In fact, according to the following reasons, we believe that iminium cations are probably to be adsorbed on the Au nanoparticles during the reaction.

First of all, *trans*-stereoselective alkylation was observed in the case of several substituted cyclic tertiary amines (Table 3). When using 4-hydroxymethyl-1-methylpiperidine as the substrate, the *trans*- α -methylene-alkynylated product was

stereoselectively obtained with the alcoholic hydroxy group intact (**3da**). Even in the presence of cyclic- α -methine C–H bonds with ester groups, cyclic- α -methylene selective alkynylation occurred to produce the corresponding *trans*-isomer stereoselectively (**3oa**). Likewise, the cyclic- α -methylene selective alkynylation of nicotine proceeded *trans*-stereoselectively (**3pa**). The *trans*-stereoselectivity is possibly derived from the attack of the alkynyl species from the opposite side of the catalyst surface to the iminium cation adsorbed on the Au nanoparticles.

Second, by referring to the previous report (*Catal. Sci. Technol.* **2021**, *11*, 3333–3346), we estimated the adsorption energy of the iminium cation of **1a** on the Au nanoparticles by DFT calculation using an Au₂₀ cluster model with no charge (M06/SDD for Au, 6-31G(d,p) for the other elements) (Supplementary Fig. 32). Although three types of initial adsorbed structures were investigated (adsorption site A, B, or C of Au₂₀ cluster shown in Supplementary Fig. 32a), all the structures converged to the adsorbed structure at the site C as shown in Supplementary Fig. 32b. The adsorption energy was calculated to be 11.6 kcal/mol based on Gibbs energies of Au₂₀ cluster, the corresponding iminium cation of **1a**, and Au₂₀ cluster adsorbed by the corresponding iminium cation of **1a**. Thus, the iminium cation suggested to be adsorbed and stabilized on the Au nanoparticle.

Third, when the oxidation of **1a** in the absence of **2a** was carried out using Au/HAP with/without ZnBr₂, the conversion of **1a** to the corresponding iminium cation, enamine and amide was quite low; however, the presence of **2a** drastically improved the conversion of **1a**, especially with ZnBr₂ (Supplementary Table 6), suggesting that the nucleophilic addition of **2a** removed the iminium cations (and enamines) adsorbed on Au nanoparticles to increase the turnover number of amine oxidation.

Therefore, it is proposed that the nucleophilic addition of the alkyne promoted by the Zn species to the iminium cation adsorbed on the Au nanoparticle affords the desired propargylic amine.

Table 3. Tertiary amine substrate scope of the combined catalytic system comprising Au/HAP and Zn species.^a

^aReaction conditions: **1** (0.3 mmol), **2a** (0.6 mmol), Au/HAP (100 mg, Au: 2.5 mol%), ZnBr₂ (13 mg, 20 mol%), toluene (2 mL), 95 °C, O₂ (1 atm), 24 h. Isolated yields are shown. ^bAu/HAP (160 mg, 4 mol%). ^c**1** (1 mmol), **2a** (0.5 mmol), ZnBr₂ (11 mg, 10 mol%). ^dPhCF₃ (2 mL). ^eGC yield. ^fAu/HAP (200 mg). ^g**2a** (1.8 mmol), MS-4A (300 mg). The ratios of *cis/trans*-isomers were determined by ¹H NMR analysis or isolated yields." (see page 32, Table 3)

“ (a) Optimized structures of Au₂₀ and iminium cation of 1a

(b) Optimized structure of Au₂₀ adsorbed by the iminium cation of 1a

Supplementary Fig. 32 Optimized structures of (a) Au₂₀ cluster with no charge, the corresponding iminium cation of **1a**, and (b) Au₂₀ cluster adsorbed by the corresponding iminium cation of **1a** based on DFT calculation. The Au₂₀ cluster model was constructed by referring to our previous report^{S12}. Although three types of initial adsorbed structures were investigated (adsorption site A, B, or C of Au₂₀ cluster shown in this figure (a)), all the structures converged to the adsorbed structure at the site C as shown in this figure (b). The adsorption energy was calculated based on Gibbs energies of Au₂₀ cluster, the corresponding iminium cation of **1a**, and Au₂₀ cluster adsorbed by the corresponding iminium cation of **1a** in this figure.” (see SI, page 56)

Supplementary Table 6. Comparison of **1a** conversions with/without **2a** and/or ZnBr₂.^a

Entry	2a [eq.]	ZnBr ₂ [mol%]	Conv. of 1a [%]	Yield [%]	
				7a	3aa
1	0	0	18	<1	—
2	0	10	26	<1	—
3	1	0	36	<1	18
4	1	10	77	<1	57

^aReaction conditions: **1a** (0.5 mmol), **2a** (0 or 0.5 mmol), Au/HAP (Au: 1.5 mol%), ZnBr₂ (0 or 10 mol%), PhCF₃ (2 mL), 95 °C, O₂ (1 atm), 20 h. Conversions and yields were determined by gas chromatography using biphenyl as an internal standard.” (see SI, page 29)

Furthermore, we also indirectly proved the existence of an iminium cation as the reaction intermediate through kinetic isotope effect (KIE) investigation using **2a-d** for the alkylation of **1a** in the presence of Au/HAP and ZnBr₂. The H/D exchange ratio of **2a-d** was estimated by GC-MS during the reaction, revealing the quick deuterium depletion of **2a-d** (estimated deuteration ratios as of 10, 20, 30, and 40 min: ~17, 14, 10, and 11%, respectively) (Supplementary Figs. 23 and 24). On the other hand, surprisingly, **3aa** was deuterated in ~62%, 57%, 58%, and 56% as of 10, 20, 30, and 40 min after the reaction started using **2a-d**, respectively (Supplementary Figs. 23 and 25). Moreover, **1a** was not deuterated at all during the reaction (Supplementary Fig. 26). To reveal the deuterated position of **3aa**, after the reaction for 24 h, **3aa** was isolated via column chromatography. NMR analyses of the isolated product revealed that 1-methyl-2-(phenylethynyl)piperidine-3,3-*d*₂ (**3aa-β-d**₂) was obtained selectively (deuteration ratio: 13%). Thus, these results indicated that enamines formed from iminium cations accepted deuterium instead of proton to convert into deuterated iminium cations during the reaction (Supplementary Fig. 27). This is an evidence of iminium cation presence in the catalytic system, and irreversible concerted one-proton/two-electron transfer to O₂ form iminium cations was strongly supported by the regiospecifically β-deuterated **3aa** formation. Azomethine ylide formation and isomerization of iminium cations were also excluded in

this catalytic system. The deuteration ratio decrease of **3aa** after the reaction for 24 h (deuteration ratio: 13%) compared with the initially formed **3aa** in the reaction (deuteration ratio as of 10 min: ~62%) is probably derived from increase of proton source (e.g. H₂O) as the oxidation reaction proceeds. In other words, the addition of D₂O is assumed to give deuterated propargylic amines from non-deuterated amines and alkynes via deuteration of the iminium cation intermediate. In fact, in the presence of a large amount of D₂O (3 mmol, three equivalents to **1a**), selectively β-deuterated propargylic amine **3aa-β-d₂** (deuteration ratio: 76%) was successfully synthesized in 44% yield (Supplementary Fig. 28), which will be beneficial in the synthesis of deuterated medicines.

Supplementary Fig. 23 Kinetic isotope effect investigation on α-alkynylation of **1a** in the presence of Au/HAP and ZnBr₂ using **2a** or **2a-d** as an alkyne substrate, respectively. Reaction conditions are indicated in this figure, and the yields were determined by GC using biphenyl as internal standard. GC-MS spectra were recorded in scan mode, and deuteration ratios of each sampling were approximately estimated from the relative intensity of $m/z = 103$ based on those of **2a** and **2a-d** (D: 98%) and from the relative intensity of $m/z = 200$ based on those of **3aa** and **3aa-d₂** (D: 13% and 76%).” (see SI, page 47)

Supplementary Fig. 24 GC-MS patterns of **2a** obtained by scan mode for $m/z = 101-105$ to detect the deuterium scrambling using **2a-d** as an alkyne substrate under the reaction conditions indicated in Supplementary Fig. 23: (a) **2a** and (b) **2a-d** (D: 98% determined by $^1\text{H NMR}$). (c) 10 min, (d) 20 min, (e) 30 min, and (f) 40 min after the reaction started.” (see SI, page 48)

Supplementary Fig. 25 GC-MS patterns of **3aa** obtained by scan mode for $m/z = 197$ – 203 to detect the deuterium scrambling using **2a-d** as an alkyne substrate or D_2O as an additive under the reaction conditions indicated in Supplementary Fig. 23 or 28: (a) **3aa**, (b) **3aa- β - d_2** (D: 13% determined by 1H NMR) isolated 24 h after the reaction using **2a-d** as an alkyne substrate, (c) **3aa- β - d_2** (D: 76% determined by 1H NMR) isolated 25 h after the reaction using D_2O as an additive. (d) 10 min, (e) 20 min, (f) 30 min, and (g) 40 min after the reaction using **2a-d** as an alkyne substrate started.” (see SI, page 49)

Supplementary Fig. 26 GC-MS patterns of **1a** obtained by selected ion monitoring mode for $m/z = 97-103$ to detect the deuterium scrambling using **2a-d** as an alkyne substrate under the reaction conditions indicated in Supplementary Fig. 23: (a) **1a**. (b) 10 min, (c) 20 min, (d) 30 min, and (e) 40 min after the reaction started.” (see SI, page 50)

“

Supplementary Fig. 27 Proposed mechanism of the corresponding iminium cations and enamine formation by aerobic oxidation of **1a** in the presence of Au nanoparticle catalysts.” (see SI, page 51)

Supplementary Fig. 28 (a) β-Deuterated propargylic amine synthesis. Reaction conditions are indicated in this figure, and the yield was determined by isolation. The deuteration ratio was determined by ¹H NMR. (b) ¹H NMR, (c) ²H NMR, and (d) ¹³C NMR spectra of the isolated 3aa-β-d₂” (see SI, page 52)

On the basis of the discussion mentioned above, we added the aforementioned figures/tables and the following sentences to the revised manuscript and SI.

“Even in the presence of cyclic- α -methine C–H bonds with ester groups, cyclic- α -methylene selective alkynylation occurred to produce the corresponding *trans*-isomer stereoselectively (**30a**). Likewise, the cyclic- α -methylene selective alkynylation of nicotine proceeded *trans*-stereoselectively (**3pa**), and as mentioned above, **3da** was also obtained as the *trans*-isomer stereoselectively. The *trans*-stereoselectivity is possibly derived from the attack of the alkynyl species from the opposite side of the catalyst surface to the iminium cation adsorbed on the Au nanoparticles.” (see page 13, lines 11–17)

“When using a radical clock as a substrate (**1s**), which could be expected to undergo ring opening upon oxidation via HAT and not to undergo ring opening by the iminium cation formation from density functional theory (DFT) calculations (Supplementary Figs. 12a and 13), for the alkynylation with **2a**, the corresponding alkynylated products (**3sa**) were obtained in moderate yields (diastereomeric ratio = 75:25) without formation of any ring-opened product (Fig. 2a).” (see page 15, lines 1–6)

“By contrast, when **2a-d** was used as the substrate instead of **2a**, the production rate of **3aa** decreased a little ($k_H/k_D = 1.3$) although GC-MS patterns revealed the quick H/D exchange of **2a-d** (estimated deuteration ratio as of 10 min: ~17%) (Supplementary Figs. 23 and 24). Considering the aforementioned kinetic analysis and the quick H/D exchange, this decrease of **3aa** production rate was not probably derived from KIE of **2a** C–H cleavage. On the other hand, surprisingly, **3aa** was deuterated in ~62% as of 10 min after the reaction started using **2a-d** without any deuteration of **1a** (Supplementary Figs. 23, 25 and 26). After the reaction for 24 h, the product was selectively isolated via column chromatography as 1-methyl-2-(phenylethynyl)piperidine-3,3- d_2 (**3aa- β - d_2**) (deuteration ratio: 13%). Thus, these results indicated that enamines formed from iminium cations accepted deuterium instead of proton to convert into deuterated iminium cations during the reaction (Supplementary Fig. 27). This is an evidence of iminium cation presence in the catalytic system, and irreversible concerted one-proton/two-electron transfer to O₂ form iminium cations was strongly supported by the regiospecifically β -deuterated **3aa** formation. Moreover, azomethine ylide formation^{34,35,68} and isomerization of iminium cations⁶⁹ were excluded in this catalytic system. The deuteration ratio decrease of **3aa** after the reaction for 24 h compared with the initially formed **3aa** in the reaction is probably derived from increase of proton source (e.g. H₂O) as the oxidation reaction proceeds. In other words, the addition of D₂O is

assumed to give deuterated propargylic amines from non-deuterated amines and alkynes. In fact, in the presence of a large amount of D₂O (3 mmol, three equivalents to **1a**), selectively β -deuterated propargylic amine **3aa- β -d₂** (deuteration ratio: 76%) was successfully synthesised in 44% yield (Supplementary Fig. 28), which will be beneficial in the synthesis of deuterated medicines⁷⁰.” (see page 17, line 18, page 18, lines 1–18, page 19, lines 1 and 2)

“Then, the desired propargylic amine is produced via the nucleophilic addition of the alkyne to the iminium cation. The *trans*-stereoselectivity to alkynylation of substituted cyclic tertiary amines is thought to be derived from this step possibly due to adsorption of the iminium cations on the Au nanoparticle. In fact, DFT calculation results using an Au₂₀ cluster model and the corresponding iminium cation of **1a** indicated that the iminium cation is adsorbed and stabilized on the Au nanoparticle (Supplementary Fig. 32). In addition, when the oxidation of **1a** in the absence of **2a** was carried out using Au/HAP with/without ZnBr₂, the conversion of **1a** to the corresponding iminium cation, enamine and amide was quite low; however, the presence of **2a** drastically improved the conversion of **1a**, especially with ZnBr₂ (Supplementary Table 6), suggesting that the nucleophilic addition of **2a** removed the iminium cations (and enamines) adsorbed on Au nanoparticles to increase the turnover number of amine oxidation. Thus, the nucleophilic addition of the alkyne promoted by the Zn species to the iminium cation adsorbed on the Au nanoparticle presumably affords the desired propargylic amine in this step. Finally, the Au–OOH species accepts a proton to afford Au, H₂O and O₂, closing the catalytic cycle.” (see page 20, lines 9–18, page 21, lines 1–4)

“68. D. Seidel. The redox-A³ reaction. *Org. Chem. Front.* **1**, 426–429 (2014).

69. Zheng, Q.-H., Meng, W., Jiang, G.-J. & Yu, Z.-X. CuI-catalyzed C1-alkynylation of tetrahydroisoquinolines (THIQs) by A³ reaction with tunable iminium ions. *Org. Lett.* **15**, 5928–5931 (2013).

70. Pirali, T., Serafini, M., Cargnin, S. & Genazzani, A. A. Applications of deuterium in medicinal chemistry. *J. Med. Chem.* **62**, 5276–5297 (2019).” (see page 29, references 68–70)

“**DFT Calculations:** All calculations were performed using the Gaussian 16 Rev B.01 or Rev. C software^{S1}. Geometry optimizations and single-point energy calculations were conducted using the B3LYP functional^{S2,S3} or M06^{S4} functional with SDD^{S5} basis sets for Au and 6-31G(d,p)^{S6} basis sets for the other elements. For the spin multiplicity, all structures were calculated as the singlet state. All thermodynamic data were calculated at the standard state (25 °C and 1 atm).” (see SI, page 4)

“S4. Zhao, Y. & Truhlar, D. G. The M06 suite of density functionals for main group thermochemistry, thermochemical kinetics, noncovalent interactions, excited states, and transition elements: two new functionals and systematic testing of four M06-class functionals and 12 other functionals. *Theor. Chem. Acc.* **120**, 215–241 (2008).

S5. Andrae, D.; Häußermann, U.; Dolg, M.; Stoll, H. & Preuß, H. Energy-adjusted ab initio pseudopotentials for the second and third row transition elements. *Theor. Chim. Acta* **77**, 123–141 (1990).

S6 Hehre, W. J.; Ditchfield, R. & Pople, J. A. Self-consistent molecular orbital methods. XII. Further extensions of gaussian-type basis sets for use in molecular orbital studies of organic molecules. *J. Chem. Phys.* **56**, 2257–2261 (1972).” (see SI, page 57)

“S12. Miyazaki, R., Jin, X., Yoshii, D., Yatabe, T., Yabe, T., Mizuno, N., Yamaguchi, K. & Hasegawa, J. Mechanistic study of C–H bond activation by O₂ on negatively charged Au clusters: α,β -dehydrogenation of 1-methyl-4-piperidone by supported Au catalysts. *Catal. Sci. Technol.* **11**, 3333–3346 (2021).” (see SI, page 58)

Comments

b) Among the isotope effect studies presented herein, I believe that the $k_H/k_D = 1.2$ presented Supplementary Fig. 20 may not be valid, as D-labelled terminal alkynes often undergo deuterium depletion in the presence of Au NPs.

Responses

In the previous version, the synthesized phenylacetylene-*d* (**2a-d**, deuteration ratio: 94%) was contaminated by a small amount of CH₃CN used as the synthetic solvent (**2a-d** purity: 98wt% determined by ¹H NMR), which possibly caused the $k_H/k_D = 1.2$. Thus, by referring to another report (*J. Org. Chem.* **2011**, *76*, 8394–8405), we carried out the synthesis of **2a-d** via H/D exchange of **2a** in D₂O without using CH₃CN followed by the extraction of **2a-d** using dichloromethane, affording the desired **2a-d** (120 mg) without contamination (deuteration ratio: 98%, determined by ¹H NMR). By utilizing the purely synthesized **2a-d**, we carried out kinetic isotope effect (KIE) investigation on α -alkynylation of **1a** in the presence of Au/HAP and ZnBr₂. As a result, the similar $k_H/k_D = 1.3$ was observed (Supplementary Fig. 23), indicating that the contamination of CH₃CN was irrelevant to the previously observed $k_H/k_D = 1.2$.

As mentioned in the responses to your comment (a), the H/D exchange ratio of **2a-d** was estimated by GC-MS during the reaction, revealing the quick deuterium depletion of **2a-d** (estimated deuteration ratios as of 10, 20, 30, and 40 min: ~17, 14, 10, and 11%, respectively) (Supplementary Figs. 23 and 24). Considering the quick H/D exchange and the kinetic analysis indicating that the turnover-limiting step is the amine

oxidation step with O₂ on the Au nanoparticles, this decrease of **3aa** production rate was not probably derived from KIE of **2a** C–H cleavage. In addition, although selectively β-deuterated propargylic amine **3aa-β-d₂** (deuteration ratio: 76%) was successfully synthesized in the presence of a large amount of D₂O (3 mmol, three equivalents to **1a**) (Supplementary Fig. 28), the yield was lower than that of **3aa** without D₂O. To reveal the reason for the decrease of **3aa** yield in the presence of D₂O, either H₂O or D₂O was added to the present alkylation, revealing the inhibition effect of water on **3aa** production possibly because of water adsorption on Au nanoparticles (Supplementary Fig. 29). Furthermore, the $k_{\text{H}_2\text{O}}/k_{\text{D}_2\text{O}} = 1.5$ was observed without much deuteration of **2a** (~6–9%) and almost equivalent to that of using **2a** or **2a-d** ($k_{\text{H}}/k_{\text{D}} = 1.3$). Thus, the regeneration step of Au–OOH species to Au species might be affected by deuteration of **2a** or addition of D₂O.

“

Supplementary Fig. 29 Kinetic isotope effect investigation on α -alkynylation of **1a** in the presence of Au/HAP and ZnBr₂ using **2a** as an alkyne substrate with H₂O or D₂O, respectively. Reaction conditions are indicated in this figure, and the yields were determined by GC using biphenyl as internal standard. GC-MS spectra were recorded in scan or selected ion monitoring mode, and deuteration ratios of each sampling were approximately estimated from the relative intensity of $m/z = 103$ or 200.” (see SI, page 53)

Therefore, we added the aforementioned Supplementary Fig. 29 and the following sentences to the revised manuscript and SI.

“To reveal the reason for the decrease of **3aa** yield in the presence of D₂O, either H₂O or D₂O was added to the present alkynylation, revealing the inhibition effect of water on **3aa** production possibly because of water adsorption on Au nanoparticles (Supplementary Fig. 29). Furthermore, the $k_{\text{H}_2\text{O}}/k_{\text{D}_2\text{O}} = 1.5$ was observed without much deuteration of **2a** (~6–9%) and almost equivalent to that using **2a** or **2a-d** ($k_{\text{H}}/k_{\text{D}} = 1.3$). Thus, the regeneration step of Au–OOH species to Au species might be affected by deuteration of **2a** or addition of D₂O.” (see page 19, lines 3–8)

“**Synthesis of phenylacetylene-d (2a-d)**: In the following manner referred to the previous report,^{S11} phenylacetylene-d (**2a-d**) was synthesized. Into a Pyrex glass test tube, phenylacetylene (3 mmol) and D₂O (5 mL) were added. After purging the air in the test tube with Ar, the mixture was stirred at room temperature for about 3 days. The resulting mixture was extracted by dichloromethane three times and dried over Na₂SO₄. After removing Na₂SO₄ by filtration, evaporation of the solvents gave the desired **2a-d** (120 mg) (deuteration ratio: 98%, determined by ¹H NMR).” (see SI, page 8)

“S11. Alonso, F., Moglie, Y., Radivoy, G. & Yus, M. Multicomponent click synthesis of 1,2,3-triazoles from epoxides in water catalyzed by copper nanoparticles on activated carbon. *J. Org. Chem.* **76**, 8394–8405 (2011).” (see SI, page 58)

Comments

Additionally, I am quite skeptical regarding the information gained from the intermolecular KIE of $k_{\text{H}}/k_{\text{D}} = 3.6$, between **1a** and **1a-d₄** suggesting the α -methylene C–H bond cleavage the turnover-limiting step. As more possibly a stable iminium cation bound on Au NP is formed as an intermediate, I would anticipate a $k_{\text{H}}/k_{\text{D}} > 1$ regardless of the kinetic profile of the alkynylation step.

Responses

We assume that you mentioned the intermolecular KIE using **1a** and **1a-d₄** in the same reactor, which is called Experiment B in the famous essay written by Eric M. Simmons and John F. Hartwig (*Angew. Chem. Int. Ed.* **2012**, *51*, 3066–3072). Considering the competitive adsorption of iminium cations on Au nanoparticles and the selectivity determining amine oxidation step, as you mentioned, the coexistence of **1a** and **1a-d₄** could afford KIE > 1 whatever the turnover-limiting step is. However, in the case

of this study, the intermolecular KIE of 3.6 between **1a** and **1a-d₄** was determined from independently calculated production rates of **3aa** and **3aa-d₃** in separate reactors (Supplementary Fig. 22), which is called Experiment A in the aforementioned essay, without any concerns about competitive oxidation or adsorption. Thus, we believe that the turnover-limiting step was correctly determined as the α -methylene C–H bond cleavage of tertiary amines from the KIE.

To clarify the method of KIE measurement, we revised the manuscript as follows.

“Moreover, we examined the kinetic isotope effects (KIEs) using 1-methylpiperidine-2,2,2,2-*d*₄ (**1a-d₄**) or phenylacetylene-*d* (**2a-d**). The intermolecular KIE between **1a** and **1a-d₄** based on independently determined production rates was large ($k_{\text{H}}/k_{\text{D}} = 3.6$), revealing the α -methylene C–H bond cleavage of **1a** as the turnover-limiting step (Fig. 2e, Supplementary Fig. 22)⁶⁵. The kinetic results show that the turnover-limiting step is the amine oxidation with O₂ on the Au nanoparticles, which is an evidence of the concerted one-proton/two-electron transfer from adsorbed amines to adsorbed O₂.” (see page 17, lines 9–15)

“65. Simmons, E. M. & Hartwig, J. F. On the interpretation of deuterium kinetic isotope effects in C–H bond functionalizations by transition-metal complexes. *Angew. Chem. Int. Ed.* **51**, 3066–3072 (2012).” (see page 29, reference 65)

“Fig. 2 Overview of the α -methylene specific alkyne addition mechanism. a, Alkyne addition of radical clocks (1s** or **1t**) with **2a**. Isolated yields are shown. b, Scrambling of deuterium using **1a-d₂**, **1a** and NaBD₄ or **1a** and D₂O. c, Alkyne addition of **1a** with **2a** under Ar. GC yields are shown. d, Dependence of the **3aa** production rate on the concentration of **1a**, the concentration of **2a** and the O₂ partial pressure. e, Kinetic isotope effects on the alkyne addition using **1a** or **1a-d₄** with **2a**. f, Plausible mechanism of the α -methylene-specific alkyne addition. d.r. = diastereomeric ratio. n.d. = not detected.” (see page 22, Fig. 2)**

Comments

Based on the analysis provided above, publication in Nature Chemistry is recommended after significant improvement of the mechanistic analysis.

Responses

As shown in the aforementioned responses, the revised manuscript, and the revised SI, the mechanistic studies were quite improved, and we believe that this revised version will be accepted as an Article of *Nature Communications*.

<To Reviewer #3>

Comments

In my opinion, it is an excellent, well written paper. Everything is very well explained and clearly stated. It can be accepted as it is.

In terms of personal preference, I would prefer seeing some more figures in the paper, not have almost everything in SI, however that is a matter of the author's preference.

Responses

Thank you for your kind review and highly evaluating this paper. Although you said that this paper could be accepted as it was, according to the editor and the other reviewers' comments, we revised the manuscript and the Supplementary Information, especially about the introduction, substrate scopes, and mechanistic studies. We believe that the revised version is more suitable for the publication in *Nature Communications*. Please confirm the revised manuscript and Supplementary Information.

REVIEWERS' COMMENTS

Reviewer #1 (Remarks to the Author):

The authors have revised the manuscript and given detailed responses according to the requirements and comments of the reviewer. The substrate scope has been expanded by exploring regio- and diastereoselectivities for unsymmetric tertiary amines, making the work stronger. Although several works have reported the functionalization of tertiary amines, this manuscript provides a general example of an α -methylene C–H alkylation of tertiary amines. The reviewer thinks the current version of manuscript meets the standard for publication in Nature Commun..

Reviewer #2 (Remarks to the Author):

The authors have significantly improved the quality of this manuscript by performing additional experiments and theoretical calculations as suggested. Publication is now recommended.